# Sustainable and Bio-Based Food Packaging: A Review on Past and Current Design Innovations

**DOI:** 10.3390/foods12051057

**Published:** 2023-03-02

**Authors:** Florencia Versino, Florencia Ortega, Yuliana Monroy, Sandra Rivero, Olivia Valeria López, María Alejandra García

**Affiliations:** 1Centro de Investigación y Desarrollo en Criotecnología de Alimentos (CIDCA), UNLP-CONICET-CICPBA, 47 y 116, La Plata 1900, Argentina; 2Facultad de Ingeniería, Universidad Nacional de La Plata (UNLP), 47 y 115, La Plata 1900, Argentina; 3Facultad de Ciencias Exactas, Universidad Nacional de La Plata (UNLP), 47 y 115, La Plata 1900, Argentina; 4Planta Piloto de Ingeniería Química (PLAPIQUI), UNS-CONICET, Camino La Carrindanga km.7, Bahía Blanca 8000, Argentina

**Keywords:** barrier properties, active material, intelligent packaging, circular economy, biobased inks and dyes, bioadhesives, bioplastic, agri-food by-products

## Abstract

Food loss and waste occur for many reasons, from crop processing to household leftovers. Even though some waste generation is unavoidable, a considerable amount is due to supply chain inefficiencies and damage during transport and handling. Packaging design and materials innovations represent real opportunities to reduce food waste within the supply chain. Besides, changes in people’s lifestyles have increased the demand for high-quality, fresh, minimally processed, and ready-to-eat food products with extended shelf-life, that need to meet strict and constantly renewed food safety regulations. In this regard, accurate monitoring of food quality and spoilage is necessary to diminish both health hazards and food waste. Thus, this work provides an overview of the most recent advances in the investigation and development of food packaging materials and design with the aim to improve food chain sustainability. Enhanced barrier and surface properties as well as active materials for food conservation are reviewed. Likewise, the function, importance, current availability, and future trends of intelligent and smart packaging systems are presented, especially considering biobased sensor development by 3D printing technology. In addition, driving factors affecting fully biobased packaging design and materials development and production are discussed, considering byproducts and waste minimization and revalorization, recyclability, biodegradability, and other possible ends-of-life and their impact on product/package system sustainability.

## 1. Introduction

Food packaging comprises items, such as containers, cups, tableware, straws, bags, wraps, and boxes, that protect or contain food. Within a growing urbanized population food packaging is needed for the transport, storage, and consumption of food products. By 2050 the world’s population is estimated to reach 9.7 billion, two-thirds of which will be living in urban areas with the consequent increase in food requirements and changes in food-consumption patterns [1,2]. People living in cities tend to lead more hectic lifestyles, spending little time buying or preparing food, preferring processed food to fresh foods, and purchasing mainly in supermarkets or convenience stores to save time. Such patterns are clearly evidenced nowadays in high-income countries, though small shops and traditional grocery stores are going out of business and being replaced by supermarkets even in developing countries, thus leading to lager supply chains strictly dependent on packaged food [2,3]. 

A large portion of food is lost along the supply chain due to damage caused by different environmental factors, such as moisture, oxidation, thermal degradation, or microbial contamination [4]. On average, 30% of the edible part of the global food production is lost or wasted in the supply chain [2]. Thus, several research and policies have focused on reducing such losses [5,6,7]. Bishop et al. reported that, only considering UK households, over 2 million tons of fruit and vegetables waste are expected to be generated every year by 2030, along with 105.7 kilo tons of plastic waste from the packaging in which is purchased [8]. Therefore, not only does an increase in food demands entail greater land-use requirements, soil degradation, biodiversity loss, eutrophication, freshwater scarcity, energy resources depletion, and overall climate change by the agri-food industry, but also more plastic-waste pollution from agriculture and food packaging. Furthermore, a large portion of food packaging is discarded with food waste, making it difficult and expensive to separate in waste management systems, ending up in less preferred waste treatments and incurring an environmental burden across both food and plastic value chains [8,9].

Most food packaging is directly disposed of after use (95%) and more than a third do not enter the recollection systems [10]. In May 2018, the European Commission approved a directive by which single-use plastics were prohibited or at least restricted, intending to reduce plastic waste pollution of seas and oceans [11]. The set of approved standards comprises requirements for product labeling, waste collection, and public awareness regarding the environmental problem and responsible consumption. Since then, a slight reduction in fossil-based plastic and a mild increase in biobased plastic and post-consumer recycled plastic production has been reported in Europe, though the overall global plastic production has continued to grow (reaching 390.7 million tons in 2021) [12]. Aiming to develop more sustainable food packaging, some significant efforts have been focused on biobased and/or biodegradable materials, especially bioplastics, paper, and cardboard. Alternative cellulose sources to wood have been studied aiming at tree and biodiversity conservation in forests and rainforest ecosystems [13,14,15,16]. Plant-based bioplastics, both biodegradable (such as poly-lactic acid (PLA), polyhydroxybutyrate (PHB), or biopolymer-based plastics) and non-degradable (biobased polyolefins: e.g., bio-PE, bio-PP, and bio-PET), are currently being commercialized and agri-food industry by-products and waste are being studied as raw materials to minimize agriculture environmental impact and land-use [15,17,18,19,20,21,22,23,24]. However, enhanced recyclability capacity, retrieval quantity, and good separation through waste management systems are crucial for non-biodegradable bioplastics to be effectively recycled through conventional recycling infrastructure and technology [25]. Compostable bioplastics may be preferred to non-degradable plastics for single-use applications, such as food packaging and foodservice ware, when reutilization or reduction is not possible and recyclability is limited [26,27]. Nonetheless, even for this type of material, adequate waste stream management and industrial composting facilities are needed, since compostability will only occur under specific conditions [27,28,29]. Several studies have been conducted on the environmental impact of biobased materials over fossil conventional plastic packaging [8,28,30,31,32,33,34,35,36,37,38,39,40,41,42], yet no general result has been reported on which is the more sustainable alternative. The overall environmental impact of the package depends on the packed food product, distribution logistic and distances, clean energy resources, available waste management systems, re-use frequency, recyclability and/or biodegradability, end-of-life scenario selected for the packaging material, and overall cost of production, use, and disposal [27,43,44,45]. 

Regarding the further reduction of food waste and losses, two main strategies are followed. On the one hand, there is agri-food industry by-products reduction and revalorization as feedstock for biomaterials and active compounds recovery, aiming to reduce by-products treatment costs and overall environmental impact [46,47]. On the other hand, the development of active and intelligent packaging to prevent food spoilage and increase food safety is being performed [4,40]. Extended shelf-life increases the chances of the food to be eaten with preserved nutritional value and reduced toxicity risk. Besides, the use of sensors and new identification technologies may provide better detection of food degradation markers to optimize food distribution and conservation and thus reduce food loss [48]. 

The food packaging industry is a thriving market that plays a key role in the modern economy. Food packaging design is crucial to guarantee food safety throughout the supply chain, optimize storage and transportation, and facilitate the communication of product information to retailers and customers [49]. Approximately 9% of the vast amount of research reported on food packaging (41,602 publications from 1960–2022) focuses on packaging design. More sustainable food packaging design aims for minimum environmental burden at the end-of-life of the food product and package. Consequently, research and innovation in food packaging are growing fast. The literature review within the last decade indicates an accelerated increase in annual publications involving sustainable food packaging (19–35% annual increase) together with active and intelligent food packaging (13–33% annual increase). Among food, packaging paper is the second (34%) most used and studied material after plastic (37% both rigid and flexible) [50]. A sustained increase in work related to innovations and innovative food packaging is observed, with an incipient but fast increase in studies on biobased, especially bioplastic food packaging, and active and intelligent packaging for food, both showing a remarkable growth in the last couple of years (Figure 1). Such tendencies evidence the work towards developing environmentally friendly and sustainable food packaging material focusing on: (i) enhancing biopolymers performance through physical, chemical, and enzymatic modification and composite formulation, (ii) developing natural-based nanocomposites for active and intelligent sustainable materials, (iii) synthesizing new biobased and biodegradable materials, (iv) scaling up bioplastic processes, (v) searching for new renewable sources, especially undervalued raw materials, and (vi) improving materials reutilization and recycling capacity [17,23,24,47,51,52,53,54,55,56,57,58,59,60,61]. 

Novel packaging systems are aimed not only at ensuring food safety and traceability, but have also gained great importance for building more sustainable food chains, reducing food losses and waste, as well as the overall packaging environmental impact. In previous works, a recompilation of the latest research and development on biobased packaging materials and composites form agri-food waste and byproduct was conducted, with a special focus on materials properties and processing technologies for greener production [55,61,62]. Thus, this review gathers the advances reported in the last decade in the research and development on packaging materials and design aiming to improve food chain sustainability within a circular economy paradigm. Biobased materials with enhanced barrier and surface properties, and specific activity for food conservation, such as antimicrobial or antifog properties, were reviewed. An overview on recent advances in intelligent packaging development, especially biobased sensor development by 3D printing technology, is provided. A further assessment of fully biobased packaging design and materials, considering waste minimization and revalorization, recyclability, biodegradability, and compatibility of the packaged food products with current waste management technologies is given. 

## 2. Sustainable Food Packaging Design

Considering that over 80% of the environmental impact of a product is determined at the design stage, design plays a key role in its sustainability [63]. In food packaging, an efficient design can increase the sustainability of the packaging and the food product it contains by minimizing both food and packaging waste at their end-of-life. However, ensuring food preservation accounts for a larger portion of the overall life-cycle environmental impact of the packed food product. On average, the packaging is estimated to comprise only 10% of the energy input required for a single person’s weekly food consumption and can ensure that the residual 90% is not wasted throughout the supply chain [64]. The relative environmental impact that packaging has on a food product depends on the food type, mainly on its perishability, cost, and overall emissions and resources depletion in production. For example, greenhouse gasses (GHGs) emission from dairy products and meats is greater than those from fruits and vegetables by 13–18%, largely exceeding GHGs in packaging manufacturing and end-of-life [65]. Therefore, focus on minimizing food waste for animal food products would yield greater benefits to the system, while emphasis should be placed on reducing the impact of the packaging of fruit and vegetables [66]. Life cycle assessment (LCA) of both the food product and the preferred packaging system should be evaluated in each case for better decision-making, considering real available waste management systems and possible optimized alternatives. Consequently, apart from the protection of the product, designing the most effective and sustainable packaging is a complex process involving numerous sectors implicated in the whole supply chain, including the intended market [28,64]. 

To minimize the environmental impact of the packaging system, it is important to consider its capacity to contain, protect, and preserve the product as to extend its shelf-life and guarantee food safety, but it should also be adequately sized, easy-to-open and easy-to-empty, and with clearly accessible information to prevent food from being wasted. In addition, the packaging materials should meet the desired mechanical and barrier properties remaining as light-weighted as possible, food-safe, ideally reusable or recyclable, and disposed of with minimum to zero pollution. Several research efforts have focused on enhancing materials barrier properties for sustainable food packaging (See Section 3).

Most of the design decisions are aimed to reduce packaging and food waste, and all of them indirectly have an economic impact. A large portion of domestically wasted food could be because the packaging does not meet consumer needs, such as packages that are too large or difficult to empty [65,66]. For instance, easily emptied packaging systems reduce both product waste and cost per unit and make cleansing easier for better recyclability of the packaging material, therefore lowering water use for this matter. There are various research and novel start-ups commercializing easy emptying packaging coating, especially biobased and food-safe materials [67,68,69]. In this regard, LiquidGlide^®^, a spin-off from the Varanasi Lab at MIT (Massachusetts Institute of Technology), has launched a novel, biobased, durable, superhydrophobic material for packaging applications.

Additional packaging attributes that can influence food waste comprise resealability, easy-to-open, grip, portioning or dosage, and communication of food safety/freshness information [70,71]. Innovations in intelligent and smart food packaging (See Section 4.2 and Section 5.2) aim to facilitate detection, recording, and communication of food quality and safety to overcome problems in expiration date labeling lack of uniformity within the global market that can lead to confusion in the actual state of the product [66]. Unclear or wrong date labeling may therefore cause potential food intoxication in consumers or unnecessary disposal of still fresh products, increasing human health risks and food waste. Nonetheless, further emphasis on the LCA need of these packaging/product systems is required to assure net environmental benefits, focusing on intelligent and smart systems cost, reusability, and recyclability [66,72].

It is important to identify the packaging functions that influence food waste to achieve more efficient packaging systems. Good mechanical resistance helps protect the product from damage due to vibrations during transport on rough roads and compression during storage as provides protection from contamination to guarantee food safety [70]. Besides, damaged products may also be rejected by customers. A real understanding of the natural characteristics and shelf-life of the food is needed to select the best packaging system. Verghese et al. reported significant annual losses of bananas in Australia partially because of improper packaging, which could be significantly overcome by replacing reusable plastic crates with more stable corrugated carton crates and clear bags for fruit neck protection and further in situ retailer owned ripening rooms [64]. Similarly, a novel non-toxic biobased *aloe vera* coating on papaya fruit has proven to delay ripening and significantly extend their shelf-life [73].

Materials with barrier properties have been widely studied to reduce food perishability (See Section 3). Depending on the food matrix characteristics, the diffusion or transmission of different substances, e.g., oxygen, carbon dioxide, ethylene, nitrogen, water vapor, and volatile molecules, microorganisms, light, or heat will need to be controlled to avoid undesired chemical and physical reactions in the food product that could lead to unpleasant odors or flavors, or even spoilage. Multi-layered packaging systems have the potential to provide barrier properties to multiple undesired or hazardous compounds simultaneously or provide the barrier materials with the mechanical or thermal resistance needed, and reversely. Currently, multi-layered packaging is widely used in the food and beverage industry, such as juice and milk cartons, usually consisting of 75% paperboard, 20% of plastic (mostly PE), and 5% of aluminum foil [74]. Despite their potential for increasing food shelf-life and mostly entailing less material to achieve the same functionality as mono-materials, their recyclability is rather difficult [75,76]. Palombini et al. studied the sustainability of packaging used for organic food in Brazil, showing that the dominant features in poor or non-recyclable polymer packaging were metallization and the use of opaque materials, polymer blends, the presence of adhesives for labeling or multi-layered packaging, and printed labeling over the packaging [76]. Better sorting of disposed multi-layer packaging would be required for high-quality recycled materials, though these are currently found in different recycling pathways, hampering their recyclability. Kiaser et al. clearly overview the two main available technologies for multi-layered packaging recycling: compatibilization methods (for polymer-based packaging) and component separation by dissolution–reprecipitation technique or delamination for independent recycling of each component when possible [77]. The major limitations of these methods are that: (i) common compatibilization does not seem to be feasible for post-consumer packaging due to fluctuation in composition and the need for more rigorous labeling and sorting of packaging waste stream; (ii) even though the dissolution–reprecipitation method could be available for the recycling of existing materials in the near future, it is a very high energy demand process; (iii) systematic delamination could be an ecologically and economically more sustainable solution if separation and adequate sorting of the delaminated layers are possible [77]. In this regard, further investigation of the optimization of multi-layered packaging recycling routes is needed. Alternatively, fully biodegradable multi-layer packaging systems designed to be composted at their end-of-life could probably reduce food waste and at the same time both avoid problems in recyclability and reduce the environmental impact of this type of food packaging. Yet, more research is needed considering LCA of the proposed materials, contemplating that a large proportion of disposed packaging labeled as compostable or biodegradable is not correctly treated. Misdirection occurs either because these materials are thrown away without proper classification or do not even enter the urban waste treatment streams, e.g., non-biodegradable plastic packaging, or are rejected from recycling plant due to a lack of available processing technologies or a shortage in volume needed for cost effective recycling, or are separated in composting facilities by mistake to avoid compost contamination or because there is no adequate separation technology to tell one type of plastic from another when the material is mixed with other biowaste [27,78,79]. 

Furthermore, novel composite and nanocomposite materials have also been studied to enhance materials barrier properties for food packaging. A focus on more sustainable materials fully biobased and biodegradable especially aiming to use by-products and waste from the agri-food industry is extensively reported in the literature review [14,17,47,55,56,57,59,60,62,80,81] (See Section 3 and Section 5.1). For instance, Báscón-Villegas et al. recently published a work on composite PLA and polybutylene adipate-co-terephthalate (PBAT) with lignocellulose nanofibers obtained from wheat straw [82]. Similarly, Silva dos Santos et al. formulated a coating paperboard packaging based on chitosan, palmitic acid, and active carbon that provides increased fat and moisture barrier for paper packaging preserving biodegradability and recyclability [83]. Besides, many new active properties can be imparted to packaging materials to extend food shelf-life [4,84,85]. For example, Manfredi et al. highlighted that milk waste savings within the fresh milk supply chain by the application of an antimicrobial coating on regular multi-layer paperboard packaging is greater than the impact generated by the coating production [86]. Active packaging contains compounds are able to preserve the organoleptic or sensory characteristics to ensure the food product’s quality. Of particular interest are active packaging materials containing natural antioxidants and antimicrobials, which not only extend the shelf-life of packaged products by preventing rancidity reactions, but also prevent the growth of foodborne pathogens [55,87,88]. Biobased nanocomposites have been studied for innovation in sensors for the better detection of food degradation markers and thus a reduction in food loss. In this regard, active and intelligent packaging is currently receiving great interest, although they are still being researched and are not widely commercialized. A thorough revision of recent research on these materials has been conducted in the following Section 4 and Section 5.2.

The development of advanced methods for recycling and remanufacturing should be considered in designing new circular and sustainable systems. Some key aspects related include recyclable and non-hazardous materials; efficient logistic and transportation systems; energy and resource efficient technologies; and product design aimed at recyclability, remanufacturability, and reusability [89]. Even though the goal of reuse and recycling is entailed within the circular economy model, this is poorly achieved for food packaging since it is difficult to guarantee the quality and safety of materials for food contact from the available waste management systems. Currently, reuse is only viable for refillable and cleanable containers (e.g., glass bottles, stainless-steel containers) [75]. In the long term, chemical safety of recycled food packaging could be achieved by replacing unsafe substances from materials design, avoiding their entrance to the recycling stream. However, a period of 10–30 years is estimated for a significant reduction in contaminants levels [75]. In the short term, clear labeling of packaging materials, consumers’ education to identify more sustainable packaging alternatives and more efficient safety and sorting systems within the waste management and recycling systems could contribute to a more sustainable use of food packaging [71,75]. Besides, sustainable recycling systems need to be cost effective in terms of the selected recycling technology, material quality of recycled products (depending on the type and conditions of secondary raw materials recollected from waste stream), availability (in terms of economy of scale), and recollection and sorting cost [28,90,91]. Facilitating recyclability has not only environmental and economic benefits, but also a social contribution, considering that dry waste sorting and selling constitute the income source for low-income workers in some countries [76]. In addition, packaging design can considerably affect the logistics in transportation, handling, and storage throughout the supply chain [92,93]. Such decisions have direct implications in distribution times and product preservation. However, the best packaging systems depends on the frequency of reuse, transportation routes and distances, and end-of-life treatment [41,64,93,94,95,96]. As regards resources reduction, water and energy saving technologies, by-products and biowaste revalorization are being achieved by the development of new biobased packaging materials and novel heat insulating materials that can help reduce energy in preserving thermosensitive foods as well as reducing their spoilage rate [17,55,56,62,97,98,99,100]. 

To reduce the environmental impact of biodegradable and compostable bioplastics, the preferred route of disposal and treatment is composting. However, most packaging materials nowadays reported as compostable (according to current legislation) require specific composting conditions (thermophilic: ~58 °C) to be fully and safely composted that are only feasible in industrial composting facilities [27,29,78,101]. Therefore, compostable bioplastic-based food packaging faces some obstacles to fulfill its purpose. A large volume of bioplastics does not enter composting systems due to the lack of industrial composting facilities or the poor sorting of these materials. Waste misdirection results from improper labeling, workers inexperience with biodegradable plastics treatment, a lack of appropriate sorting technologies in sorting facilities, or a lack of consumer education on waste separation or environmental protection. When biodegradable plastics wrongly end up in landfills, they can decompose anaerobically producing methane, a more harmful GHG than CO_2_, hindering their sustainability [28,79]. By occasional littering or waste mistreatment, bioplastics can also enter water streams and marine environments in which they are not able to biodegrade or can degrade, posing a hazard to life in such ecosystems. Research has been conducted to prove biodegradable materials’ ecotoxicity [29,101,102,103,104]. Zimmermann et al. reported that many bioplastics available on the market are just as toxic as conventional plastics, presenting clear bioluminescence inhibition of *Aliivibrio fischeri* [101]. Wang et al. thoroughly revised the available studies on plastics degradation in marine environments, highlighting that only a few works consider the toxicity of degradation products, including material fragments or debris and additives, which should not be toxic [29]. In this regard, the chemical safety of materials can be further achieved by bottom-up green chemistry design in the development of new biobased and biodegradable materials. On this matter, ecotoxicity assays of new biodegradable materials research is being conducted, especially after compost or soil degradation, usually showing promising results [103,105,106,107]. Furthermore, non-biodegradable and fossil-based plastics, biodegradable bioplastics can break into small particles, producing microplastic pollution and affecting different species of plant and animals adversely [29,102,108]. This points out the necessity of an in-depth understanding of the environmental degradability performance of available and upcoming biodegradable plastics. 

In product-packaging design, it is also important to consider and understand the consumers perception and acceptance of the packaging. Some authors imply that consumers, tend to have a poor understanding of the benefits of packaging and mostly a negative image of its role to preserve food, e.g., removing the package prematurely to let the food “breathe” [64]. On the contrary, other research findings indicate that consumers recognize the importance of packaging plays in food safety and quality in relation to the information that it provides and show a pro-environmental consciousness [109]. A study carried out by Harpen et al. using an immersive 3D virtual supermarket environment revealed that customers tend to prefer unpacked fruits and vegetables [110]. Herbes et al. indicated that consumers mostly focus on the end-of-life stage of packaging and are mostly unaware of the negative impacts of packaging in the upstream and middle-stream supply chain [111]. Consumers’ desire to pay for green or sustainable products grows as environmental awareness rises, yet the higher price of these products limits sales to a niche market [28,91]. Consequently, future research should contemplate consumers involved in packaging alternatives design as well as cost reduction for the increased accessibility of novel sustainable food packaging.

A systemic focus on understanding the relationship between packaging, products, supply chain networks, and physical processes is required in an era of multi-product physical supply chains that have a global footprint [95]. Padhi et al. identified sustainable design and development products, strategic commercialization, technology optimization, and sustainable product returns and recycling as the most important sustainable supply chain processes [96]. The conception of a new eco-design approach gives place to further innovation in packaging and products with a critical rethinking of the system and new perspectives towards sustainable production. From the literature review, some basic principles for sustainable design can be outlined [55,95,96,112]:A focus on real demand and problems and try to find solutions with social, environmental, and economic benefits altogether.A shift in the design dynamics from the application requirements up, thinking in terms of functions and services rather than the product itself.Consideration of realistic and updated life-cycle and process thinking, having an integrated view of the supply chain, thus taking into account the product manufacturing, distribution, consumption and end-of-life.Inclusion of users, stakeholders, and different experts in the design process as much as possible.Research and innovation need to be grounded on justifiable priorities within the available time frame and scope of the project.

A holistic understanding of the supply chain drivers, barriers, and opportunities is required in view of sustainable long-lasting packaging-product systems designed towards a circular economy.

## 3. Strategies to Improve Barrier and Hydrophobicity Properties for Food Packaging

With the main premise of preserving the food matrix quality, packaging industries and the scientific sector are striving to develop new materials to minimize deteriorative changes mainly due to physical and chemical modifications experienced by foods during their distribution and storage [113]. Food packaging is considered an integral part of the preservation system and, therefore, the package provides a barrier between the food matrix and the external environment, protecting it against physical, chemical, and biological damage [114]. To control the chemical and physical reactions in the matrix, packaging material must be able to act as a barrier to external factors that affect the food quality, such as the gaseous atmosphere, water activity, light, and temperature. In this sense, the barrier properties of a material are related to the protection of the food matrix inside the package by controlling mass transfer [115]. The controlled transference of diverse compounds, such as gasses, water vapor, and volatile molecules, is relevant to minimize the reactions that cause food degradation by creating a favorable atmosphere around the packed product [116]. Food stability is characterized by its chemically unstable nature and hence requires to be protected from spoilage, lipid deterioration, and microbial contamination. Therefore, polymeric materials must prevent the penetration of compounds from the surrounding environment to guarantee a “high barrier” [117]. A schematic representation of how packaging protects food from external agents, such as gaseous compounds, water vapor, UV radiation, and microorganisms, is shown in Figure 2. In this regard, some examples of alternatives to develop materials with enhanced barrier properties are also addressed. 

As it is well known that the most common degradation processes of packaged foods are caused by oxygen, such as lipid oxidation, microorganism growth, enzymatic browning reactions, and vitamin loss, among others. According to Zabihzadeh-Khajavi et al. controlling the oxygen permeability of the packaging system can extend the shelf-life of the packed food [118]. Ethylene, on the other hand, is considered the aging hormone of plants since it is responsible for the growth and ripening of fruits [119]. A very high concentration of ethylene around the food can accelerate its decay. Therefore, the elimination of this gas from the packaging or the control of its permeability is relevant to increase the shelf-life of fresh products [120]. While carbon dioxide is one of the most employed components in the gas mixtures for modified atmosphere packaging (MAP) due to its antibacterial properties [121], this gas can prevent the microbial growth of fresh meat, cheese, or baked goods, and minimize the respiration rate of fruits and vegetables [120]. Moreover, moisture content alters the nutritional and organoleptic properties and safety of food products. The reaction rates of lipid oxidation, microbial growth, and browning are altered when the food moisture content changes. Effective packaging can play a major role in maintaining product moisture to extend the shelf-life of food. 

Given the importance of quantifying the barrier properties, mainly to gasses and water vapor, it is relevant to use normalized or standardized methods that allow the comparison with other materials. As it was stressed by Baschetti and Minelli, numerous methodologies have been employed to study the permeability of different gasses and volatile compounds through polymeric matrices under diverse experimental conditions [120]. The most important international standards are American Standards and Table 1 summarizes the ASTM norms for the determination of gasses and water vapor permeability in polymers for food packaging [120,122]. Another important barrier property of food packaging is its capacity to block UV (200–400 nm) and visible (400–700 nm) radiation. The exposure of food products to UV light (natural or artificial) throughout their useful life (harvesting, elaboration, storage, distribution, sale, and preparation by consumers) can induce photo-oxidation and photo-degradation reactions that affect the quality of food [123]. Many photosensitive food components (proteins, vitamins, pigments, fats, and oils, etc.) are the substrates for these reactions in which free radicals and oxygen reactive compounds are produced, inducing food deterioration by the development of undesirable flavors and aromas, color loss, and a decrease of the nutritional value [124,125]. As an alternative to avoid photodegradation, different methodologies can be used, such as reflective layers, as well as the coating or incorporation of organic and inorganic absorbers to the polymeric matrices [115]. On the other hand, the study of the UV barrier capacity of food packages is relevant if the products, once packaged, will be submitted to microbial decontamination by light-based technologies, such as UV radiation. According to Cassar et al., this methodology can reduce the food microbial charge but also leads to product deterioration or the generation of undesirable residues [126]. Regarding visible radiation, transparent packaging allows consumers to visualize the packaged product and highlight some of the food organoleptic characteristics that condition the acceptability of the product. For this reason, opacity is a relevant optical property that conditions the quality of packaging films [127].

In accordance with other authors, it is relevant to find multifunctional high-performance barrier materials considering several issues, such as optical and mechanical properties, renewability, and bio-friendliness, to develop food packaging that assures food quality and fits all consumers’ necessities [128,129,130].

### 3.1. Enhanced Gas and Water Vapor Barrier Properties

The permeation of low molecular weight gasses/vapors through films is measured by the ability of the polymer matrix to absorb and diffuse the penetrant. The main mechanisms that affect mass-transfer of substances across packaging materials are diffusivity, solubility, and permeability, which are closely related to the composition and structure of the polymer matrix [131]. Factors, such as the shape and size of the permeant, morphology, crystallinity, and chain orientation of the polymer, influence the diffusion and solubility coefficient. Other variables that modify the matter transfer coefficient are related to the processing methods, which can alter polymeric chain configuration or induce crystallization/orientation, including polymer blending, multi-layer coextrusion, casting, or applied nanotechnology [132]. 

The enhancement in the barrier properties in nanocomposite materials is attributed to the more tortuous path created by the presence of different nanofillers. This fact is explained considering that the nanofillers force the low molecular weight molecules to adopt particular tortuous pathways, producing a significant lag time. A higher aspect ratio of fillers compels the permeating gas molecule to follow a more tortuous path, leading to improved barrier properties [133]. Furthermore, gas diffusion through materials is also controlled by the crystal structure domains. It is generally assumed that ordered crystalline domains should act as an effective barrier to the diffusion of gasses and small molecules, making the amorphous phase the only pathway available for permeation. Moreover, penetrants cannot sorb in crystalline domains because their solubility coefficients are lower compared to those of their amorphous counterparts [117].

To achieve a better barrier performance, the assembly of nanoparticles to obtain nanocomposites constitute a strategy to improve barrier properties. Organic nanoparticles, such as carbon nanotubes, nanocrystals of cellulose or starch, and inorganic nanoparticles, such as nanoclays or montmorillonite (MMT), can physically and chemically interact with polymeric matrices to induce stronger and reinforced structures, enhancing both the barrier and mechanical properties [134]. MMT nano-clays can enhance the polymer barrier performance because of their exfoliated structures, small particle size, high aspect ratio, and exceptionally large surface areas that enlarge the tortuous path for small diffusing molecules [133]. Thus, the nanomaterials can be applied to improve the performance of conventional materials due to their particle size and their large surface area [135]. 

Other approaches have been tested to regulate the transport of water/gasses through the packaging, e.g., the use of different polymer mixtures, polymer crosslinking by chemical reactions to narrow the intermediate chains, assembly of micro and nanoparticles to obtain composites, nanostructured matrices formed by an electrodynamic process, and coatings or multi-layer films to design packaging materials with high water/gas barrier properties [115,136,137]. In this regard, multi-layer packaging constitutes an emerging technique that integrates the characteristics of different polymers or layers to create a package with enhanced performance in terms of properties [77]. These materials have improved properties as compared to typical single-layer films, exhibit high-barrier to water vapor and gasses, such as oxygen, carbon dioxide, and aromatics, as well as high mechanical strength and good sealing capacity [138]. 

Multi-layer films can be prepared by different methods, such as melt coextrusion, film-forming solution molded by casting, and compression molding of sandwich structures [132]. In the design of multi-layer film packaging different layers can be assembled. In this sense, to enhance the techno-functional properties of polymers, an inner barrier layer film usually consists of polymers with higher oxygen barrier properties, while polymers with higher water vapor barrier and resistance from a mechanical point of view act as an outer layer. The barrier layer is in direct contact with the external environment, acting as a barrier to substances that cause the degradation of packaged food, such as moisture, oxygen, and microorganisms [131].

According to Alias et al., multi-layered films based on biomass combined with synthetic biodegradable polymer films obtained from natural monomers showed better transparency, water solubility, and mechanical properties than single-layer films [138]. Meanwhile, multi-layer films based on biomass sources demonstrate better barrier properties in terms of water vapor permeability and offer advanced properties in terms of oxygen permeability, still exhibiting a significant improvement in terms of physical and mechanical properties [138].

In addition, to achieve high-performance biodegradable multi-layer films with tailor-made properties, knowledge of their microstructure and film processing steps is required. The challenge is to improve the protective barrier of the individual layers and therefore optimize the design of the matrices, so current research must be redirected in this way [132]. It is pertinent to emphasize that the most important factor is the adhesion between layers to combine different layers that improve the barrier protection of the individual layers and thus optimize the resulting system. In this sense, research considering the scale, interaction, and architecture of structural layers in multi-layer matrices and their influences on barrier capacity performance is hardly reported in the literature. 

The layer-by-layer assembly generates a stratified structure with a significantly enhanced gas barrier and mechanical or even optical properties due to confinement and/or interfacial effects. Layer-by-layer assembly is based on different interactions, such as electrostatic attraction, hydrogen bonding, hydrophobic attraction, or entanglement between polymeric chains in neighboring layers. The interactions that are established depend on the characteristics of the polymers that make up the multi-layer system. Particularly, for some semi-crystalline polymers, the confinement generated by the stratified structure leads to a specific orientation of the crystals, nanoparticles, or compounds, inducing an increase in the tortuosity for diffusion [131,139].

### 3.2. Enhanced Hydrophobicity

Many biodegradable and renewable biopolymers are hydrophilic since they are conformed by polar molecules. This inherent hydrophilicity often results in moisture absorption that leads to the deterioration of the mechanical properties of the material and affects its dimensional stability, both undesirable effects for packaging applications [140].

To reduce the hydrophilicity of biodegradable and natural polymers, several methodologies can be carried out. One of these alternatives is to modify the chemical structure of the biopolymers to improve their hydrophobicity. According to this, Wang et al. stressed that chemical modifications of starch may be carried out to achieve adequate physicochemical characteristics by blocking or adding functional groups, improving the hydrophobicity of starch-based materials [52]. For instance, Petronilho et al. have worked on hydrophobic starch-based films by transesterification with sunflower oil in alkaline medium, showing increased water resistance (lower solubility and water vapor transmission, and higher contact angles) [137]. In the case of cellulose, for instance, chemical modification by silylation, esterification, amidation, and grafting are some of the reactions that can be used to obtain more hydrophobic cellulose [141]. 

Plasma treatment is another methodology usually used to increase the hydrophobicity of natural polymers. Pankaj et al. highlighted plasma treatment as a novel, more environmentally friendly technology to obtain more hydrophobic starch derivatives [142]. The effects of plasma occur on the surface of the material without altering the properties of the bulk involving the generation of reactive species, such as ions, radicals, electrons, photons, and other excited species. Generally, the plasma induces different reactions, such as surface cleansing, removal of organic contaminants, degradation (etching), cross-linking of polymer chains, and modification of the functional groups present on the surface. The different physical and chemical changes that the surface experiences depend on the gas used to generate the plasma [143]. 

Goiana et al. studied how a dielectric barrier discharge (DBD) plasma treatment affects the hydrophilicity, WVP, and mechanical properties of corn starch-based films [144]. The authors concluded that this treatment produces more hydrophobic starch materials with improved mechanical performance, equally interesting properties for food packaging. Following this idea, the use of cold plasma treatment increases the surface roughness of biopolymer films [145]. The magnitude of the effect on the film’s roughness depends on some factors, such as power supply (voltage), exposure time, and uniformity of the exposed energy of the plasma species onto the film surface [146]. Among the effects that induce cold plasma treatment, the reduction of the contact angle between the film’s surface and the water drops can be mentioned. Despite plasma treatments representing an effective method to reduce the hydrophilic character of natural polymers, they can reduce the biodegradability of materials [147]. 

A more innovative method to enhance the biopolymers hydrophobicity is based on the reduction of the contact angle between water droplets and the material surface considering its topographical features. Therefore, materials with a specific surface topography can include air compartments that hold water droplets, minimizing their contact with the material surface. In this regard, bioinspired hierarchical surfaces are very interesting mostly because of their wetting behavior properties [148]. For example, lotus leaves and rose petals present superhydrophobic surfaces with different wetting behaviors [67,149,150]. Two methodologies can be employed to obtain materials with the hierarchical structures of lotus leaves or rose petals: directly molding from hierarchical templates or growing nanowires or nanorods onto designed microstructures [148]. Following the first methodology, Luís et al. proposed a method to mimic the lotus leaf surface to fabricate zein-based films [151]. The authors demonstrated that films produced using the lotus negative template presented lotus-leaf-like rugosities, resulting in very hydrophobic surfaces. Likewise, de Oliveira Gama et al. “decorated” the surfaces of films based on thermoplastic starch (TPS) and blends of TPS/low-density polyethylene (LDPE) to improve their hydrophilic properties [152]. These authors employed micro- and nanocomponents to decorate the materials surfaces and they demonstrated the feasibility of creating topographical patterns to generate hydrophobic and even superhydrophobic features that can be useful in minimizing the water absorption.

Other techniques, such as electrospinning, have been used in the design of superhydrophobic nanostructure fibers and surfaces. Pardo-Figuerez et al. developed an innovative PLA and nanostructured silica (SiO_2_) superhydrophobic multi-layer material with PET by electrospinning and electrospraying [68]. Moreover, Zhang et al. prepared a fully biobased thermo-resistant edible super-hydrophobic coating from coffee lignin and beeswax with a promising application for food packaging of food that require high-temperature sterilization [69]. Likewise, fully biobased materials from food industry waste with enhanced hydrophobic properties were developed and studied by Gonçalves et al. [153]. The authors used recovered starch, oil, and wax from potato chips manufacturing to produce starch-based films with enhanced flexibility and wettability with tailored roughness. For a further and thorough overview of the most recent studies on the application of superhydrophobic coatings and surfaces for sustainable food packaging, the authors recommend the work of Ruzi et al. [154].

### 3.3. Enhanced UV-Light Barrier Characteristics

A strategy to enhance UV-light barrier properties includes the assembly of light stabilizers into the film matrix to deactivate the reactive degradation elements, preventing their occurrence by consuming the products responsible for initiating the deterioration reactions and hindering free radicals spread. Aluminum foil represents the best material for UV-visible light blocking, but it poses some disadvantages related to its recyclability, high cost, and non-transparency. In this sense, the smart selection and application of UV absorbers to functionalize materials for specific functions constitute a promising strategy [123].

Inorganic materials are based mainly on the metal oxide particles, such as titanium dioxide (TiO_2_), cerium oxide (CeO_2_), iron oxide (Fe_2_O_3_), zirconium dioxide (ZrO_2_), and zinc oxide (ZnO), that impart enhanced light barrier properties by scattering the incident light [123,155]. However, there are concerns regarding the diffusion of metal nanoparticles from film packaging to the food matrix and to the human body after ingestion. For this reason, the European Union legislation (Commission Regulation, 2011 (EU) N° 10/2011 on Plastic Materials and Articles Intended to Come into Contact with Food) established the type and quantity of additives in materials intended to remain in direct contact with food, though further advances in research are needed to find safe alternatives [156].

According to Song et al., inorganic nanoparticles can easily agglomerate, having a negative influence on the barrier UV performance of matrices [59]. This fact limited their large-scale industrial production and consequently the practical applications of UV-blocking films. In contrast, organic UV-blocking alternatives are identified as phenolic-type UV-absorbers which exhibit good photostability, because the compounds can absorb light energy and make it less harmful, and are typically involved with hydrogen bonds from either O–H–O bridges or O–H–N bridges [123]. Organic adsorbents are generally based on triazine, benzotriazole, benzophenone, anthranilates, dibenzoylmethane, and light amine stabilizers [157]. However, the dangers of organic absorbents for human health restrict their use by promoting the search for green alternatives. In this sense, molecules, such as flavonoids or natural compounds that act as green radiation absorbers (such as tannins, kombucha tea, lignin, grape syrup, pine extract, and others), are interesting substitutes to metal nanoparticles and organic absorbents, blocking radiation in a broad range of wavelengths and decreasing the risks associated with potential migration to the food matrices [59,157,158,159,160].

Another alternative to improve UV-light barrier characteristics is the design of a hybrid material, defined as a material that comprises different systems. In this sense, hybrid UV-blocking absorbers can result from the combination of organic UV absorbers and inorganic matrices or from the integration of inorganic UV-blocking absorbers with organic polymers [123].

## 4. Active and Intelligent Food Packaging

The increasing demand for fresh, healthy, and long-shelf-life foods require innovations in packaging design. In this sense, new, intelligent, and smart packaging able to sense and communicate information from the packaged food product has been developed. Before moving forward, it is necessary to clarify three terms that are sometimes used interchangeably: active, intelligent, and smart packaging (Figure 3). 

Active materials are specifically designed to interact with the food or its surrounding environment, modifying their composition or characteristics to preserve the organoleptic or sensory characteristics of the product to ensure its quality for longer periods of time. Antimicrobials, antioxidants, flavor and gas scavengers, and light blockers are some examples of active substances usually used in food packaging [85]. Intelligent packaging materials, on the other hand, are aimed to sense changes within the food package and provide information about the status of the food inside [161]. Although limited to detecting and communicating, intelligent packaging can enhance security, safety, and convenience, providing real-time quantitative information on package integrity and food freshness, maturity, or contamination [48,85,162]. Finally, smart packaging results from the combination of both intelligent and active packaging technology, though it is sometimes used indistinctly to refer to either one or the other [163]. However, food packaging systems with a single function of freshness keeping or monitoring may not be able to meet all practical needs [164].

The global market for active and intelligent packaging materials presents a growing tendency with a forecasted CARG of 5.5% from 2021 to 2030, estimated to reach $38.66 billion by 2030 [165], although the pandemic context has probably affected the development of this market.

### 4.1. Active Materials and Packaging Systems

Active containers, commonly used in food packaging, are those that contain a substance capable of preserving the organoleptic or sensory characteristics of a product to ensure its quality. Of particular interest are active packaging containing natural antioxidants and antimicrobials that not only extend the shelf-life of packaged products by preventing rancidity reactions, but also prevent the growth of foodborne pathogens [87]. The EU Commission defined active compounds as any substance or device that can extend the shelf-life or maintain/improve the packaging environment [166]. According to European Regulation No 450/2009, the active agent can be an individual substance or a combination of substances [167]. These compounds are included in the packaging materials formulations and exert specific functions, such as releasing or absorbing CO_2_, O_2_, ethylene, odors, flavors, antioxidants, and antimicrobials. Recently, Atta et al., Amin et al., and Pandey et al. extensively reviewed biodegradable active packaging materials for food applications [4,84,85].

Active materials can be categorized by active agent or function. There are numerous types of substances used in food packaging that can act as active agents. Thus, a classification based on the purpose of the active packaging proves clearer and more useful in terms of application. However, many active compounds can have more than one function, making their classification complex.

Active packaging materials containing food-approved synthetic active compounds were first developed. Among the synthetic antimicrobials, active films that include potassium sorbate, benzoate, and propionate in their formulation stand out [168,169,170]. Synthetic antioxidants, e.g., butylated hydroxytoluene (BHT) and butylated hydroxyanisole (BHA), have been also included in both polymeric and biodegradable film formulations and were used as active packaging to prevent lipid oxidation in foods [171].

Biobased materials have great potential to replace conventional synthetic ones for packaging applications [172,173,174,175]. The use of natural additives in food packaging formulations not only protects them from the environment, but also provides protection against microbial contamination and agents that stimulate oxidative rancidity reactions, among others, reducing the need of additives in the food matrix [4,55,84]. Some of the polymers and natural compounds can be obtained from industrial agri-food waste and their use would add value to this waste and keep it in circulation. Of particular interest is the use of chitosan or its derivatives due to intrinsic antimicrobial activity against a broad spectrum of microorganisms [176]. This biodegradable polymer derives from chitin that is mainly obtained from waste from the fishing industry, such as crustacean exoskeletons [177].

Likewise, by-products and waste from fruit and vegetable processing are an important source of bioactive compounds with high nutritional and functional value, such as vitamins, minerals, antioxidants, and antimicrobial compounds, although they are often discarded or derived for animal feed [178]. The essential oils (EOs) that can be obtained from these sources have been widely studied as additives for the development of active food packaging, mainly due to their antioxidant and antimicrobial capacity and their GRAS (generally recognized as safe) character [88,179]. Therefore, the use of active compounds derived from agricultural by-products not only contributes to the recovery of these compounds with specific activities, but also generates added value for them. For example, Bof et al. have developed and characterized active biodegradable films based on corn starch and chitosan with the addition of lemon essential oil (LEO) and grapefruit seed extracts (GSE) [180]. The inclusion of these active compounds, which are by-products or derivatives of citrus processing residues, did not affect the mechanical properties of the material and provided antimicrobial capacity on contact. Similarly, Kanmani and Rhim developed antimicrobial active films with GSE in a carrageenan matrix, with additional UV-barrier capacity, particularly important for UV-sensitive food packaging [181]. Moreover, Bof et al. have demonstrated that the biodegradable films based on corn starch and chitosan and the active film containing GSE reduced the post-harvest weight loss, without the incidence of rottenness, of packed blueberries during their refrigerated storage in comparison to clamshell PET containers [182]. Besides, their performance under real conditions of transport and commercialization was evaluated, also considering the costs and possible scaling of the process. Additionally, several EOs have been included in film formulations to impart antioxidant properties, such as rosemary extracts, tea polyphenols, eugenol, oregano, thyme extracts, red propolis, and green coffee oil extracts [183,184,185,186,187,188,189,190]. Thus, EOs exhibited combined and synergic actions since they can confer both antimicrobial and antioxidant capacities. This effect was demonstrated for gelatin films containing red propolis extract which exhibited both antioxidant capacity and inhibited the growth of *S. aureus*, *L. monocytogenes*, *S. enteriditis*, and *E. Coli* [186]. Similarly, chitosan films loaded with eugenol were also effective in the inhibition of *S. aureus* and *E. Coli* and presented antioxidant capacity [187].

Other natural compounds, e.g., α-tocopherol, ascorbic acid, lycopene, gallic acid and resveratrol, with proven antioxidant capacity have also been incorporated in polymer matrices to functionalize them [191,192,193,194,195,196]. Considering their action mechanism, antioxidant compounds can also be used as oxygen scavengers [197,198,199,200]. These compounds can be incorporated into the package as sachets, stickers, or embedded into the polymer matrix of the food packaging.

In addition, different strategies have been promoted to modulate the release kinetics of the active compound and thus ensure the effect of the active compound incorporated. Among them, encapsulation stands out, a technique of particular interest in the case of thermolabile compounds that will be incorporated into matrices that require processing at high temperatures. Likewise, novel methods have been developed for the design of micro and nano-systems that transport antioxidant and antimicrobial active ingredients. These structures allow the controlled release of the active components for different purposes [191]. The protection of active compounds by electrospinning or electrospraying are novel techniques with the potential to be scaled up [201]. Spray drying is a widely used technological tool to encapsulate active compounds due to its low cost and because it allows the continuous production of large amounts of product in short periods of time. However, it requires careful control of the operating conditions to avoid losses of the active compounds during the process [202].

In recent years, research reporting the formulation of active biomaterials, including metallic nanoparticles obtained by green synthesis techniques, has increased, highlighting Ag and Cu nanoparticles and ZnO nanorods [203,204,205]. In this regard, Ortega et al. have synthesized Ag nanoparticles by reduction with maltose in situ of the filmogenic suspension of starch-based biomaterials, characterizing the materials obtained and demonstrating their antimicrobial capacity [173,203]. They have also obtained nanocomposite materials based on corn starch by adding silver nanoparticles synthesized with reducing compounds present in lemon juice. These materials also exhibited the ability to inhibit the growth of fungi, yeasts, and bacteria [206]. Likewise, nanoparticles, e.g., those of TiO_2_ and montmorillonite, as well as zeolites, are efficient as ethylene scavengers and were useful to reduce fruit decay and maintain quality of fruits during postharvest storage [171,207]. Zeolites have also been used as CO_2_ scavengers in food packaging [208]. Nonetheless, even though progress has been made in nanoparticle migration studies and their cytotoxicity, it is still necessary to advance the framework legislation that regulates the use of nanocomposite active materials for their extensive use in contact with food [55,209].

Progress is still required in the design and large-scale manufacture of these active biomaterials for use in the food industry. Besides, the involved production costs should be considered since some biobased materials still require process optimization and the discovery of alternative raw materials for their production. Innovative research is being conducted on the preparation of composite materials with natural and synthetic additives using emerging technologies such as 3D printing, which could further lead to improved functionality of the obtained materials (See Section 5.2).

Future research in the field should be devoted to improving controlled release kinetics to obtain continuous release throughout the shelf-life of packaged food.

### 4.2. Intelligent and Smart Food Packaging

Intelligent packaging can non-destructively and in situ monitor the food quality, detect storage conditions, expiration date, safety diagnostics, monitor microbial growth, and determine the freshness of food [55,164,210,211]. One aspect of this technology is based on the use of interactive indicators, dye-based compounds that allow the evaluation of the chemical and microbiological quality of foods. Temperature changes during storage, oxygen concentration, or microbial activity can be monitored in real-time. According to Moradi et al., there are three main types of commercially available, intelligent, and smart packaging: time-temperature, freshness, and gas-release indicators [212]. Those of freshness include acid-base indicators based on pigments whose colors are affected by the pH of the environment [213]. Recently, special attention has been paid to the use of natural colorimetric indicators sensitive to changes in pH, including anthocyanins from different vegetable sources [163,214,215].

Sensors are also used in the design of intelligent packaging, some of which can be obtained by 3D printing [164,216]. To this end, nanomaterial-based functional ink formulations with innovative electronic designs and architectures printed on packaging materials are used to achieve various functionalities, such as radio frequency communication, time-temperature indicators, gas sensors, and freshness indicators, among others. New concepts, such as integrated printed radio frequency antennas; printed temperature sensors; and printed supercapacitors, using electronic inks, are still under development [217].

Nowadays, several intelligent systems are commercially available [218,219]. These can be classified into three types: sensors, indicators, and data carriers [220]. A sensor is a device that responds to physical or chemical stimuli measuring it or simply detecting it by some sort of signal emission. Specific sensors named biosensors have been developed to detect hazardous substances, such as foodborne pathogens and allergens, residual antibiotics, or pesticides [221]. These sensors can detect, record, and convey information relevant to biological systems. Most of the commercial biosensors are a combination of antibody-based receptor and optical transducer [222]. For example, Food Sentinel System^®^ developed by Sire Technologies Inc. (Crowthorne, UK) is an antibody-based biosensor where a membrane with immobilized antibodies is used as a part of the barcode that acts as the sensor. As the pathogens interact with antibodies, a localized dark bar is formed which renders the barcode unreadable. ToxinGuard^®^ developed by Toxin Alert (Mississauga, ON, Canada) is another system where antibodies are printed on polyethylene packaging material. The interaction between pathogens and antibodies results in the production of a fluorescent signal that indicates a pathogenic attack. Besides, gas sensors are employed for the detection of gaseous analytes, e.g., oxygen, water vapor, carbon dioxide, ethylene, etc., inside the package. Papkovsky and Dimitriev described optical oxygen sensors based on the principle of quenching or luminescence upon gaseous analyte contact [223]. The use of pH-sensitive dyes, such as methyl red and curcumin, in starch-PVA films for the detection of basic volatile amine released from rotten meat and fish have been reported [224].

On the other hand, indicators can determine the presence or concentration of another substance, or the reaction between two or more substances, by giving characteristic changes in color. Among them, freshness indicators provide information about the product quality by determining the chemical changes resulting from the microbial growth within the product. For instance, COX Technologies (Louisville, KY, USA) markets FreshTag^®^ which are colorimetric labels that indicate the production of volatile amines by fish and shellfish during storage [225]. Besides, the SensorQ^®^ indicator produced from a polymer matrix containing a solution of the bromocresol green dye was used to monitor meat and poultry freshness [226].

Likewise, time-temperature indicators (TTIs) are labels that provide a visual indication of the temperature abuse of perishable products, mostly frozen foods, during distribution and storage from the point of production to that of consumption. The operating principle of these devices is mainly based on different reactions between two or more substances triggered or accelerated by time and temperature variations that result in (usually irreversible) color changes of the indicator [161]. TTIs are based on diffusion, enzymatic, microbial, or polymer-based systems, several of which are commercially available. Diffusion-based indicators rely on either phase-transition or rheology and diffusion rate dependence of colored substances with temperature. For example, Khairunnisa et al. studied different vegetable oils and their combination to tailor their viscosity and the activation energy of the TTIs [227]. The 3M Company (Maplewood, MN, USA) has commercialized diffusion-based TTIs by the trade name 3M Monitor Mark^®^ and Freshness Check^®^ for a wide range of exposure temperatures and times. Kim et al. used isopropyl palmitate on a microporous polymer structure over a colored layer, as the fatty acid ester fills the pores the reflection index of the material decreases letting the lower layer show. Indicating that the length of the TTI device influences its application range. Enzymatic TTIs use the enzymatic hydrolysis reaction of the substrate, which is sensitive to temperature and or pH variations that can usually be identified by color change of the matrix [161,228]. Similarly, microbial TTIs use metabolites produced by microorganisms, namely yeast and lactic bacteria, under certain time and temperature conditions, generating pH changes, indicated by halochromic compounds [229,230,231,232,233]. These types of biosensors are being studied due to their sensitivity, safety and wide conditions range [228,229,230,231,233]. For instance, novel TTIs using laccase [233,234,235,236], lipase [237], and glucose oxidase [228] have been successfully developed. An example of commercial enzymatic TTI is VITSAB^®^, which is based on color change resulting from a pH drop due to controlled enzymatic hydrolysis of a lipid substrate [48]. Non-enzymatic browning reactions, such Maillard reactions, have been studied and modeled to tailor the reaction activation energy by varying the reactive concentrations or pH conditions of the TTI systems [238,239]. In addition, one of the most studied TTIs systems is based on polymerization reactions, typically poly-diacetylene (PDA) based. PDAs present thermochromism, among other chromisms, changing from blue to red upon stimulation with various sensing applications [240,241,242,243]. Polymer-based TTIs systems under the trademark Lifelines Freshness Monitor^®^ are commercialized by Lifelines Technology Inc. (Leicester, MA, USA). Recently, Suppakul et al. have developed a diffusion based TTI temperature sensitive device by combining this simple technology with a thermochromic polydiacetylene/silica nanocomposite [244]. Likewise, FreshCode^®^ (Varcode Ltd., Chicago, IL, USA) and Tempix^®^ (Tempix AB, Gävle, Sweden) are based on barcodes printed with fading inks that disappear due to temperature abuse [245]. These inks are colored compounds that produce transparent or lightly colored products by reaction, whose kinetics are regulated by time and temperature, directly indicating changes of food shelf life [161]. Besides, the integrity indicator Ageless Eye^®^ developed by Mitsubishi Gas Chemical Company (Tokyo, Japan) is an oxygen indicator tablet that turns pink when the oxygen concentration is less than 0.01% and blue when it goes beyond 0.5% [246]. Research and development in this area are constant, even more so than those related to commercially available alternatives.

The use of active and Intelligent packaging introduced new concepts and agents that required a clear updated regulatory framework on food contact materials. In fact, the lack of such a legal frame for many years led to reluctance by food packaging manufacturers to take on novel smart packaging systems which were not fully covered by the current legislation on food contact materials [220]. Regulations vary among countries, though they are mainly based on substance migration limits and toxicological properties [247]. The European approach is based on the theory that all materials should be explicitly cleared accordingly to listed hazardous substances, while in the United States substances that are not likely to cause any health problem are directly cleared (or deemed not to require regulation) on the basis of chemical or toxicological data extrapolation beneath minimal dietary exposure [248]. In addition to complying with current regulations, packaging manufacturers must also consider potential human health effects from the possible migration of contaminants, especially for intelligent devices that are placed in direct contact with food. Intelligent devices must also be properly labeled to increase consumer confidence in the safety of packaged foods as well as in the use of these technologies.

Finally, radio-frequency identification (RFID) tags provide the ability to identify, control and manage goods through the supply chain and have been successfully applied for this purpose. These are more advanced, reliable, and efficient than the conventional barcode tags for food traceability [48]. RFID tags for monitoring temperature, relative humidity, pressure, pH, and light exposure of the products are already available in the market which aid in enhancing food quality and safety. For example, Dogbone^TM^, which detects and measures temperature and moisture levels in the environment, and Circus^TM^ Tamper Loop, an NFC RFID tag that informs about authentication reordering, expiration, care, and label integration, were developed by Avery Dennison Smartrac (USA). Besides, this company developed the sensor AD-327 FCC^®^, which informs about supply chain management, inventory, and logistics [249]. According to a 2019 report by IDTechEx, the RFID market was $11.6 billion and will grow to $13 billion by 2022, which reflects the potential of this technology [249]. Although, the design of intelligent systems based on RFID technology still faces many challenges to overcome. Among them, the aspects related to the food safety, cost, reading range, multi-tag collision, multi-parameter sensors, recycling problems, security, and privacy of the RFID system should be solved [48]. Cost is an important factor to consider in intelligent and smart packaging development. If the cost is high in comparison to the product, a price rise could imply fewer sales and a consequent increase in product waste. Likewise, investments in these technologies may be too high from both economic and environmental perspectives [64]. Therefore, new, low-cost materials and methods for sensors, indicators, and data carriers are sought [250,251]. In this regard, 3D and 4D printing of biobased composite materials plays a key role, which may constitute the object of future research and development.

Intelligent packaging and collaboration in data sharing and information transparency may facilitate more synchronized supply chains with greater visibility and traceability, which can help reduce inventory and prevent food from going bad in stock by more precise demand forecast and sales data [64].

It is also imperative to study and control any possible negative effects of intelligent packaging on recycling systems. Bibi et al. remarked upon the need for recycling programs for RFID sensor tags [48]. Hence, materials used should be one of the most important considerations with the aim for low environmental impact elements. In this regard, bio-based smart packaging is a potential option that combines sustainable production and real-time monitoring of food quality to guarantee health safety, prevent food loss, and provide both economic and environmental benefits [252].

Table 2 summarizes recent novel biobased active materials and intelligent systems innovations as well as their applications in food packaging.

## 5. Biobased Packaging

One of the applications with the greatest impact on the growing demand for bioplastics is for containers and packaging, particularly biodegradable materials in food packaging. Before the COVID-19 pandemic, many multinationals, such as Danone, Coca-Cola, and Nestle, had committed to adopting bioplastics in their packaging [273]. In this sense, there are numerous investigations and developments of biodegradable and biobased materials due to their potential lower environmental impact. Numerous works have been reported and an extensive literature revision has been conducted by various researchers in the field of material sciences and the packaging industry [4,15,17,51,54,62,87,115,175,274,275]. A comparison of different biomass sources for bioplastics and biopolymers along with various processing technologies discussion can be found among available literature [55,61,113,114,276,277,278,279]. The latest works focus on new biomass sources to reduce bioplastics cost and climate impact due to extensive land use [14,18,19,22,23,24,46,47,55,56,280,281,282,283,284,285,286,287,288,289,290]. In view of more sustainable packaging systems, fully biobased packages are sought, considering all parts. Therefore, herein, a thorough overview of biobased adhesives, inks, and dyes for food packaging is presented.

### 5.1. Bioadhesives for Food Packaging

The term adhesive describes a formulation that can hold two or more specimens together. In other words, an adhesive can join materials by surface bonding (adhesion), with a bond possessing adequate internal strength (cohesion) [291,292] (Figure 4a). The materials which are to be joined are called adherents or substrates and the process of attaching one adherent to another is called bonding. The adhesives include a wide range of materials with very different specifications that produce adhesion through different mechanisms depending on the characteristics of the substrate. The effectiveness of the adhesive bonding capacity depends on several factors, which are intermolecular forces in the adhesive, wettability of the adhesive in the adherent, types of chemical bonds, functional groups involved, type of interface between the adhesive and the adherent, type of substrate, and surface tension generated [293]. Likewise, Dohr and Hirn reported that the strength of the adhesive bond depends on various physical properties of both the substrate and the adhesive [294].

Both the way the adhesive is applied and its chemical composition determine the possible applications. Other factors, such as the dispersion of the adhesive over the substrate and the contact area, which in turn have a great impact on the adhesive forces that can develop, are also relevant [294].

Although in former times adhesives were prepared from natural sources, synthetic formulations of adhesive are nowadays based on thermoset resins: phenol-formaldehyde, an alkaline catalytic salt (PF), urea-formaldehyde, an acid catalytic salt (UF), melamine-formaldehyde (MF), and polymeric diphenylmethane diisocyanate resin (pMDI) [295,296]. These synthetic adhesives cause health problems during the manufacture process. For instance, formaldehyde was declared a carcinogen by the World Health Organization (WHO) in 2004. Similarly, the International Agency for Research on Cancer (IARC) reclassified formaldehyde as carcinogen category 1 in 2004 [297]. In addition, a comparative increase in price and the foreseen lack of availability for petroleum-based synthetic adhesives have prompted the development of green derivatives from economic and renewable resources.

Biobased adhesives are formulations based on natural raw materials, which are not derived from mineral or fossil sources. This term includes adhesives formulated with biopolymers obtained from plants, animals, and natural gums [298]. In the search to develop bio-based adhesives with properties similar to traditional formulations, proteins, tannins, lignin, and polysaccharides are eco-compatible biopolymers that could fit these requirements [299,300]. Their use is mainly limited to paper, cardboard, aluminum foil, and wood for construction applications. These adhesives develop stickiness quickly but exhibit low-strength properties. Most are soluble in water and use water as a solvent agent. They are supplied as liquids or dry powders to mix with water, though some are dispersions [300].

Figure 4b shows a classification of adhesives according to the source of the polymers from which they are prepared, as it was reported by Ebnesajjad and Landrock [301].

The global market for adhesives and sealants is growing rapidly and is expected to be worth $85.8 billion by 2026, with packaging being one of the most relevant applications [302]. Adhesives in packaging industries are critical to the structure of most paper and paperboard packaging. In this regard, paper-based materials have been broadly applied as packaging material for food products.

From a production point of view, adhesive selection can significantly affect process line efficiency and production performance [303]. Adhesives are commercialized in many shapes and types, and the choice will be determined by the substrates on which the adhesive for bonding will be applied, the machinery used in the process, and other factors, such as potential requirements for food-safe materials [303].

The types of adhesives used in the paper industry are water-based adhesives, both synthetic and biopolymer-based (starch, cellulose derivatives, proteins), solvent-based adhesives (polyurethane and acrylic base), and 100% solid adhesives, such as heat-sealing adhesive and hot melts. Gadhave and Gadhave reported the development of a water-based heat-expandable adhesive that has thermally insulating properties and has been used in protective food packaging [303].

Protein-based adhesives (animal or vegetable) were the first polymers to be used in ancient times as adhesives in contact with the skin due to their biocompatibility and biodegradability properties. Likewise, protein-based adhesive formulations have been developed as substitutes for formaldehyde resins, particularly urea-formaldehyde resins in applications for construction material adhesives and paper and coating manufacturing [298].

There is numerous research on adhesives to bond wood made from tannins [304], lignin [305], cellulose derivatives [306] and modified starches [302,307,308], and vegetable proteins, such as casein [309], gluten [310], and soy proteins [311]. Adhesives based on animal protein can be formulated from collagen, gelatin, or casein. Collagen is prepared from different parts of animal bodies, such as skins or bones, and gelatin is derived from the denaturation of collagen. The reversible gel-sol conversion by water absorption is an important property for the application of these adhesives. Casein is obtained from cow’s milk by acid precipitation using different acid media, which can result in diverse properties depending on the resulting molecular structures. The primary structure of this protein can be chemically modified to achieve different rheological behaviors [292,309].

Research related to the development of biobased composite adhesives for applications in food packaging is still incipient. Fatty acids derived from vegetable oils were used as the base for monomers that once polymerized form pressure-sensitive adhesives [312]. Heinrich, Wang et al., Wu et al., Mahieu et al., and Ding et al. have reported the use of blends based on starches and proteins from different sources for green bioadhesives preparation [58,300,313,314,315]. Due to the high number of polar groups in both proteins and polysaccharides, the hydrophilicity of composite systems is often the biggest obstacle to overcome in adhesive formulations [300]. For proteins, this includes making functional groups available for crosslinking, for instance, by tertiary and quaternary structure modification. Chemical modifications can be made in the structure of the polysaccharides and proteins (starch, cellulose, gelatin, soy proteins, among others) to improve the crosslinking of the matrix using crosslinking agents. Kumar et al. have reported that the use of crosslinked starch with citric acid results in materials with good mechanical performance [316]. These properties are explained and revealed from a structural point of view by the interactions established between the carboxyl groups of citric acid and the hydroxyl groups in starch. On the other hand, Olomo informed that adhesives chemically treated using HCl as a cassava starch gelatinization modifier were of higher quality than those modified in presence of NaOH [317].

While native starch contains only hydroxyl groups and is limited in scope, chemically modified starch shows superior water resistance properties for adhesive applications [302]. Other modifications to prepare biobased adhesives with improved water resistance may involve esterification, transesterification, alkylation, acetylation, succinylation, or enzymatic reactions. The use of starch as a raw material in the manufacture of adhesives has the advantages of its renewability, biodegradability, and availability, in addition to its low cost and non-toxicity [317]. Cassava (*Manihot esculenta*) is produced mainly in the tropical and subtropical regions of Africa, South America, and Asia. The high starch content of cassava and its higher proportion of amylopectin, compared to other starch sources, make it an important source of biopolymers to be used in the development of biobased adhesives. It can be employed as native starch, but it can be modified by different means to improve its properties of consistency (viscosity) [315]. Cassava starch as a base to produce adhesives has many notable characteristics, including high paste viscosity, high paste clarity, and high freeze-thaw stability [318]. Besides, Li et al. worked on improving the adhesion-to-fibers and film properties of corn starch by starch sulfo-itaconation for a better application in warp sizing [299].

Biobased adhesives based on colloidal solutions in general dry slowly and therefore require a very long setting time. Bioadhesives containing starch, dextrin, and/or casein are primarily used in labeling applications, while starch-based formulations are also widely used in the obtention of corrugated boards [319]. Starch-based adhesives are mainly composed of water, starch, sodium hydroxide, borax or boric acid, and other additives, such as preservatives, adhesion enhancers and defoamers, among others [320]. Most starch-derived adhesives are used in the paper and textile industries as bonding agents and gluing materials. Corrugated cardboard is produced by the adhesion of a grooved layer of paper to another flat layer. To join them, a two-phase adhesive is usually employed: a liquid and a solid phase containing a mixture of starch and sodium hydroxide or native starch and borax, respectively. Developing biobased adhesive formulations for bonding different substrates, such as corrugated cardboard paper, glass, among others, to propose applications in the formation of film packaging and in the labeling process, constitute a great challenge.

Currently, most containers that are in contact with food have adhesive in their structure. Adhesive formulations can be found in three different forms (Figure 4a). It can form the structure of food packaging by combination with different materials (usually polymers, paper, cardboard, or glass), commonly known as laminate or multi-layer packaging. Moreover, the adhesive can help to provide the geometry of the container (box sealing) or can be used for labeling. The most common way to find adhesives on food packaging is of the first type; where the adhesive is applied on the total surface of the materials or substrates, joining different materials and forming multi-layer materials. Examples of practical applications are flexible film lamination, paper-film combinations, cardboard-film, aluminum-film, cardboard-aluminum, cardboard-aluminum-plastic, rigid multi-layer systems based on plastic, sacks, bags, among others. For such applications, the adhesive industry uses a wide variety of raw materials and formulations, combining different compounds to form special types of biobased adhesives.

There is currently a growing push in the packaging and adhesive industry to improve the sustainability of processes and products. Therefore, the general trend in the adhesive industry for use in food packaging is a reduction in the use of solvents as well as molecular weight components of the adhesive that could more easily migrate from the package to the food.

The use of biobased adhesives in the labeling of glass containers is an interesting alternative, since for this purpose there are few studies that contemplate the use of bioadhesives. The removal of the label-adhesive using only water is a desirable characteristic not only from an operational point of view, but also from an economic and environmental level. Easily removable labels would facilitate the sorting of recyclable waste packaging and therefore increase the recyclability of the whole package and reduce water and energy use. In addition, compared to the common industry practice for label removal using diluted NaOH solutions, water is a low-cost natural resource that, most importantly, could be reused, ultimately requiring simple effluent treatments before discharge.

Regarding the legal background surrounding adhesives intended to be used for food contact applications, normally, the adhesives are used to stick together packaging materials and not intended for direct food contact. However, the adhesives as components of the packaging material might contribute to the migration of constituents into the food matrix. Adhesives, as well as food contact materials, are regulated according to EU Framework Regulation (EC) No 1935/2004. Plastic materials and articles are additionally regulated by a specific measure, Regulation (EU) No 10/20115 on plastics and therefore harmonized at EU level. This regulation establishes, among other requirements, a list of authorized compounds. It is pertinent to emphasize that adhesives do not yet have such specific harmonized legislation. Alternatively, reference is made to the opinions of the European Authority focusing on aspects of Food Safety, Council of Europe resolutions, national legislation, and even non-European legislation for risk assessments [319]. The Commission Regulation (EC) No. 2023/2006, recommends procedures to assure the safety of adhesives for food contact applications, particularly on Good Manufacturing Practices for adhesives. Yet, there is currently no legal obligation for adhesive manufacturers to provide a declaration of conformity with Regulation (EC) No 1935/2004. However, if the adhesive falls under the Plastics Regulation, the adhesive manufacturer shall provide the specific information to enable the adhesive user to ensure compliance of substances with migration potential [319].

According to the specifications reported by Romero Zaliz et al., adhesives for returnable glass containers must be formulated with raw materials that are included in the positive lists of the FDA and the Argentine Food Code (Mercosur) for use in Food Containers and Equipment in Contact with Food (Argentine Food Code Chapter IV, FDA 21 CFR 175.105) [321]. For this reason, there is a high interest in the development of new strategies to produce new renewable materials that totally or partially replace petroleum-derived reagents, resulting in innovative products with special functionality, less toxicity, high biocompatibility, and/or biodegradability [322].

Using natural resources or bio-based materials as adhesive raw materials could help future societies become less dependent on hazardous chemicals, volatile organic compounds, and petroleum-derived chemicals; in addition to promoting safer working conditions. Consumer trends toward green products are prompting plastics industries to investigate more benign alternatives to petroleum-based polymers. Moreover, the recent classification of formaldehyde as a harmful substance has accelerated the investigation of more ecological and renewable alternatives, such as protein-based adhesives, to avoid harmful emissions both during production and during the lifetime. Furthermore, the use of bio-renewable or waste raw materials helps to reduce the carbon footprint, aligned with the current circular economy framework. As an added benefit, the inherent biodegradability of renewable materials, e.g., starch, polyhydroxyalkanoates, or cellulose, is often higher than that of synthetic materials, e.g., polypropylene and polyethylene.

The shift towards more ecofriendly alternatives has manifested itself in the adhesives industry first through the gradual change from solvent adhesives to water-based or high-solid content adhesives, and now by the renewed interest in the design of biobased adhesives. Nonetheless, it is convenient to modify the formulations based on natural polymers, such as polysaccharides and proteins, to improve their rheological properties, their adhesion capacity, and their mechanical resistance when applied for the bonding of different substrates to be used in food packaging. Consequently, composite adhesive formulations obtained from chemically modified biopolymers constitute an innovative proposal to overcome such difficulties.

Further research on the selection and compatibility of biobased adhesive formulations is essential since the adhesive characteristics presented will depend as well on the type of substrates applied to. Additionally, progress is still required in relation to the legislation that frames the adhesives that can be used in the manufacture of containers in contact with food.

### 5.2. Biobased Inks and Dyes in the Food Industry

The use of dyes and pigments dates back to 3500 BC, when the civilizations used natural extracts for coloring. Then, in 1856, the first synthetic dye was created by Perkin and a significant number of dyes were quickly discovered thereafter and adopted by industries [323,324]. Nowadays, the most used colorants (dyes, inks, and pigments) by industries are organic molecules derived from petrochemicals and other chemicals that are causing important environmental damage. According to The Synthetic Dye and Pigment Global Market Report 2022, the global synthetic dye and pigment market grew from $54.54 billion in 2021 to $59.82 billion in 2022 and is expected to grow to $79.45 billion in 2026 [325]. People are more inclined to be nature-friendly and health-conscious, which has created a revolution in research and development in eco-friendly and non-toxic colorants, pushing dyes and pigments manufacturers to shift back to natural dyes. Several countries have imposed a ban on the import of synthetic dyes. For example, the use of azo dyes is banned in India owing to their environmental and health impacts. In general, environmental considerations are becoming vital factors during the selection of consumer products.

Natural dyes are applied in several areas (Figure 5) because they have some special properties, e.g., soothing color, are biodegradable, non-hazardous, non-carcinogenic, and present antimicrobial resistance, among others [324,326,327]. In relation to the food industry, colorants are widely used since, in many products, they are a highly valued attribute by consumers. They are currently being extended to food packaging technologies by applying colorimetric indicators or sensors to exhibit color changes with variations in pH, temperature, and gas for control of the food’s quality. In the last five years, especially in food packaging, the research of natural dyes and inks has increased according to the Scopus database (November 2022), although it is not a field that has been extensively studied.

Natural colorants can be classified according to their hue (red, yellow, brown, blue, purple, black, green, and orange), their origin (vegetal, animal, bacterial, fungal, etc.) or their chemical structure as follows: flavonoid derivatives (anthocyanins), isoprenoid derivatives (carotenoids), nitrogen heterocyclic derivatives (betalains) (Table 3). According to Iqbal and Ansari, these natural products present the advantages of being eco-friendly, biodegradable, renewable, not health hazardous or non-toxic; can be obtained from waste biomass, and in some cases present antibacterial and/or UV protective properties [324]. In contrast, natural dyes are expensive because of limited available sources, they are difficult to produce, and the reproducibility of shades is hard to control, among others.

Colorants are widely used in the food industry as an essential ingredient since in some products the color is a relevant characteristic. Viera et al. established that color is responsible for 62–90% of the consumers’ acceptability [328]. There is a growing worldwide concern for food quality and safety in the modern era, and given the availability of natural colorants, their use in the food industry is increasing. In this sense, there has been extensive research on the use of natural colorants in food products [326,329,330,331].

Another use of natural colors is associated with 3D printing technology (additive manufacturing), since this technology can increase the acceptability of certain foods because of the change in the form and presentation, yielding a more attractive product for consumers. Inks for 3D food printing are classified based on their ease of use, nutrition-related components (protein, carbohydrate, fat, fiber), and functional compounds, such as vitamins and antioxidants, that can be incorporated through pigments or natural inks derived from fruits and vegetables extracts. Besides, the additives applied to edible inks play an important role in improving the flow behavior, sedimentation, and lubrication properties of the material to be printed [332]. In this sense, there are some publications regarding the use of fruits and vegetables extracts to obtain more attractive presentations for kids. Lee et al. used spinach powder and xanthan gum to print spinach dispersions and Derossi et al. printed fruit snacks using banana, beans, mushrooms, and lemon juice, while Qiu et al. investigated the 3D printing of apple and edible rose blends as a dysphagia food [333,334,335].

Escalante-Aburto et al. suggest that the production of 3D food printing must be considered a zero-waste technique to reduce the environmental impact [332]. Thus, the development of bioinks should be focused on using low-carbon and low-water footprint food ingredients, leading to the introduction of a new market for novel and edible composites. Bioinks should be sold as a packaged ingredient that is inserted into the printer to obtain 3D-printed foods. The packaging material must be recyclable or biodegradable and innocuous to fit in with the sustainable technology and food safety concepts.

On the other hand, in 2013, Skylar Tibbits put forward the term 4D printing in his TED talk based on 3D printing [336]. Such 4D printing is based on smart materials and includes a fourth dimension: time, in addition to 3D spatial coordinates. Thus, 4D printing enables the shape, properties, or function of the 3D-printed products to change over time under environmental stimuli, such as temperature, concentration differences, water, pH, or light [336].

Color changes represent one of the most common applications of 4D printing food. He et al. studied the spontaneous color change induced by pH in ready-to-eat 4D foods with anthocyanin-rich purple sweet potatoes, while Ghazal et al. aimed to investigate changes in colors and flavors of 3D-printed healthy food products in response to an external or internal pH stimulus [336,337]. For the formulation, a combination of red cabbage juice, vanillin powder, potato starch, and different fruit juices was used. The changes in color, texture, flavor, and taste induced by the stimulus were determined, revealing that the color of the 3D-printed product changed from blue (control sample) to red, purple, violet, blue-green, and green-yellow colors when sprayed with pH solutions of different pH (2–10). In addition, dried 4D samples exhibited color and anthocyanin stability when stored at room temperature for three weeks. Likewise, Chen et al. discussed the possibility of using microwaves to stimulate the color change of 3D-printed curcumin lotus root gel [338]. Based on traditional 3D printing technology, this work uses microwaves as the stimulus to obtain the color change of printed products and provides a new method for producing colorful and attractive food through 4D printing.

Another use for 4D printed materials could be as quality sensors or indicators for food packaging.

The obtention and use of natural inks or dyes provide an advantage for food packaging. Because of their low molecular weights, coloring agents, photo-initiators, solvents, and oils may migrate from printing inks, favoring natural dyes over synthetic dyes. Studies to obtain natural dyes from beet, red beet, potato, red onion, quince, black carrot, and hibiscus to use them in areas where contact with the human body is frequent, such as textile, tissue engineering, and food packaging, are being carried out [339]. In addition, it is known that these plant extracts used as inks present antimicrobial activity, and for this reason, they are preferred for active and intelligent packaging development.

There are many studies concerning the use of natural dyes or inks in the development of intelligent packaging such as films or 4D printing sensors [212,214,337,340,341]. In this sense, Alizadeh-Sani et al. have critically reviewed pH-sensitive smart packaging films based on natural colorants for the monitoring of food quality and safety [342]. Meanwhile, Tracey et al. released an advanced 3D printing approach to intelligent food packaging [164]. Among the smart packages, those containing TTIs are the most studied. As mentioned in Section 4.2, these indicators adhere to the surface of the packaging and produce an irreversible color change in response to environmental conditions. Rachmelia and Imawan developed the TTI label using black corn extract and chitosan matrix. In this study, anthocyanin was obtained from black corn extract, while chitosan was used as a matrix [343]. The use of non-toxic ingredients was aimed at making the TTI label safe to apply in packaging products. The label’s color was observed over time at temperatures of 10, 25, and 40 °C. The label changed color from purple, to blue, to yellow being the fastest color changes at higher temperatures (40 °C) and slowest at low temperatures (10 °C). Furthermore, Mataragas et al. developed a microbial TTI based on the violacein (a microbial violet pigment produced by *Janthinobacterium* sp.) formation for monitoring the shelf-life of minced beef-vacuum-packed cooked meat products. When the temperature varied from 0 to 15 °C, the color changed from purple to violet [231]. Other materials with a similar operation are pH and gas indicators, which due to these factors play an important role being indicative of food quality, shelf-life, microbial growth rate, and food deterioration [156,342,344]. Erna et al. obtained curcumin/rice starch films for sensitive detection of hypoxanthine in chicken and fish meat, since the deprotonation of curcumin occurs when the indicator is exposed to a pH level of 9 or above, where the indicator’s color changes from yellow to a reddish-brown or wine-red color [345]; while Boccalon et al. developed potato starch composite films containing red onion skin extract as intelligent pH indicators for food packaging [346]. The films were tested by monitoring their color changes when applied to meat and milk storage.

According to Tracey et al., highly sensitive, self-indicating, multifunctional smart components using biocompatible, nontoxic materials via lower cost intelligent packaging systems and devices can be developed by 3D printing in comparison to conventional fabrication methods [164]. Zhou et al. used coaxial 3D printing followed by ionic crosslinking to create fruit freshness keeping and visual monitoring labels with high pH sensitivity and effective shelf-life extension capability [347]. Cellulose nanofibers (CNF)-based ink with blueberry anthocyanin was used to create the shell of fibers, exhibiting high formability and print fidelity as well as sensitive visual pH responsiveness for freshness monitoring. Chitosan containing 1-methylcyclopropene (1-MCP), an ethylene receptor inhibitor with good freshness-keeping performance and no toxicity, was loaded into the hollow microchannels of the fibers. The 1-MCP was trapped by the electrostatic effect of chitosan and CNF exhibited a sustained release behavior [347]. Finally, the 3D-printed labels prolonged the shelf-life of litchis by approximately six days. On the other hand, Li et al. developed an interlayer with chitosan, mulberry anthocyanin as a natural dye, and lemongrass essential oils as an antibacterial agent and antioxidant using a 3D printer, and cassava starch as a protective layer to form indicator films [341]. These were used to monitor the quality, freshness, and preservation of cold meats by observing the color changes of the indicators. When chilled pork spoiled, the color of the indicator films changed from red to gray-blue (RGB), and the RGB tone values could be analyzed by a smartphone application to determine pork freshness [341].

The use of smart packaging based on natural colorants and biodegradable films is intended as an attractive alternative for food packaging due to their low or nontoxicity, eco-friendliness, easy preparation, biodegradability, availability, renewability, and pollution-free properties. The main challenges of natural inks and biodegradable food packaging are related to the improvement of their functionality at reasonable costs. Likewise, progress in the assessment of their safety is needed. According to Bautista et al., this agrees with the global challenges of the packaging industry, some of the most important being: (1) increase sustainability of manufacturing processes, (2) improve the recyclability of materials, and (3) improve performance and functionality. In addition, the printing of labels with bioinks or the use of smart inks, including natural dyes, in the design of freshness indicator devices is a field that is still in its infancy with great potential [217].

**Table 3 foods-12-01057-t003:** Classification of natural dyes according to their chemical structure and source.

Color	Classification	Source	Example	Reference
Yellow/ Orange/ Red	Curcumin	Plant/Vegetable	Turmeric (*Curcuma longa* L.)	[348]
	Carotenoids	Plant/Vegetable	Carrot (*Daucus carota* L.), Annatto (*Bixa orellana*), Tomato (*Solanum lycopersicum*), Paprika (*Capsicum annuum* L.), petals of marigold (*Tagetes erecta* L.)	[349,350]
	Aryl carotenoids	Microorganisms	*Brevibacterium linens, Streptomyces mediolani, Mycobacterium aurum*	[351]
Red/Pink	Betalains	Plant/Vegetable	Beetroot (*Beta vulgaris* L.), *Opuntia lasiacantha*	[339,349]
	Carminic acid	Microorganisms	Cochineal (*Dactylopius coccus*)	[350]
	Anthocyanins	Plant/Vegetable	*Hibiscus rosa sinensis* flowers	[352]
Blue/ Purple	Anthocyanins	Plant/Vegetable	Grapes (*Vitus labruscana* L.), red cabbage (*Brassica oleracea var. capitata f. rubra*), cherry (*Prunus cerasus*), blueberry (*Vaccinium sect. Cyanococcus*), red onion skin (*Allium cepa* L.), Beetroot (*Beta vulgaris* L.)	[323,326,350]
	Tyrian purple (6,6′-dibromoíndigo)	Animal	Mollusks *Bolinus brandaris*	[349]
	Ultramarine Blue	Mineral	Lapis lazuli	[349]
Green	Chlorophylls	Plant/Vegetable	Spinach (*Spinacia oleracea*), kiwi pomace (*Actinidiaceae*), green beans (*Phaseolus vulgaris*), grass, alfalfa (*Medicago sativa*)	[57,350]
	Terre-Verte (Green Earth)	Mineral	Mixture of hydrosilicates of Fe, Mg, Al, and K (gluconite and celadenite) but other minerals are likely to be present	[349]
	Malachite		Copper carbonate hydroxide	

## 6. Conclusions

Changes in lifestyle have directly influenced the type of food consumed, as well as consumption habits, which in turn has generated the need for the food industry to develop new containers and packaging. Today, practically any food product is marketed packaged, not only to contain the food, but also to protect it throughout the entire production chain, until it reaches the point of sale or consumption. Hence, the way we produce and consume food has substantial environmental, social, and economic impacts, requiring sustainable solutions for proper and efficient land use, better food preservation technologies during processing and packaging, and novel transport, distribution, and marketing systems to guarantee that these costs are well exceeded by benefits.

Food packaging fulfills very important functions for food preservation, protecting it from external agents, preventing physical, chemical, and/or microbiological contamination, as well as its possible adulteration. In this way, they fulfill the function of a barrier against the environment that surrounds them, protecting food from humidity, oxidation, UV radiation, and microorganisms. New packaging technologies aim to further extend the shelf-life of food products by active systems that can slow the natural oxidation process or avoid microbial growth. Intelligent and smart packaging can further monitor food quality and spoilage to guarantee food safety for the consumers and prevent food losses due to improper conservation or transportation, or inefficient logistics and marketing. Investments in packaging have the potential to reduce overall environmental impacts associated with food production, distribution, and consumption, aiming to minimize food waste.

However, systemic LCA approaches to determine the optimum product/package combination are needed. Effective assessment requires updated qualitative and quantitative information. Specialized sustainability agents within the industry, academia, and government are needed to develop a sustainability culture. Further efforts should also be made to raise awareness and educate food and packaging supply chain stakeholders and consumers on the role of packaging in extending food shelf-life and the opportunities to use it more effectively. Such opportunities entail: enhanced protection and larger shelf-life for fresh food products with potential tailored solutions; recovery of surplus and unsalable produce from farms to rescue supply chains; revalorization and use of agri-food industry byproducts and waste for packaging materials production; proper size (or portioning), function (i.e., easy-to-open, easy-to-empty, fit-for-purpose), reusable and/or recyclable food packaging design; new packaging materials and technologies to extend shelf-life (i.e., active, intelligent and smart packaging); unified labeling regulation for better use-by or best-before date indications for manufacturers, retailers, and consumers to avoid confusion regarding productions dates and prevent the unnecessary disposal of food; and intelligent packaging and data sharing to create more synchronized supply chains to reduce excess or out-of-date stock.

A gap exists between product/packaging system design, materials supply, manufacturing and commercialization, and the return flow from recyclable materials that enter the waste management stream, which hinders circularity. To close the cycle, a holistic understanding of the supply chain components, their opportunities, and limitations is required for transitioning sustainable production systems towards a circular economy.

## Figures and Tables

**Figure 1 foods-12-01057-f001:**
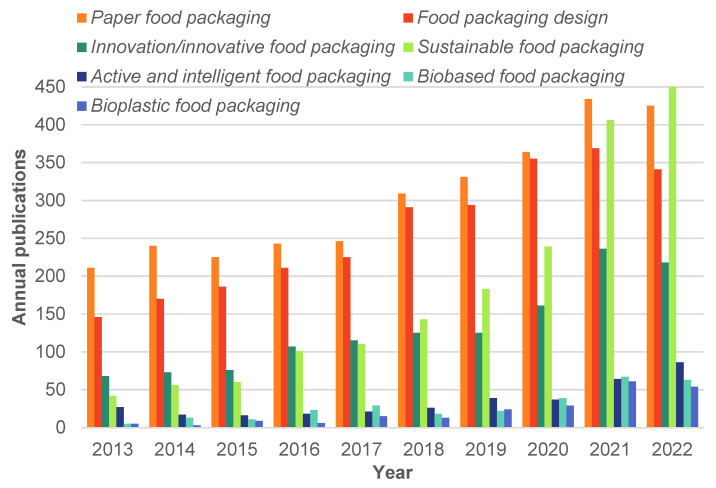
Total number of publications reported in literature in the last ten years on food packaging for specific keywords. Literature review carried out on Scopus search engine of bibliographic databases.

**Figure 2 foods-12-01057-f002:**
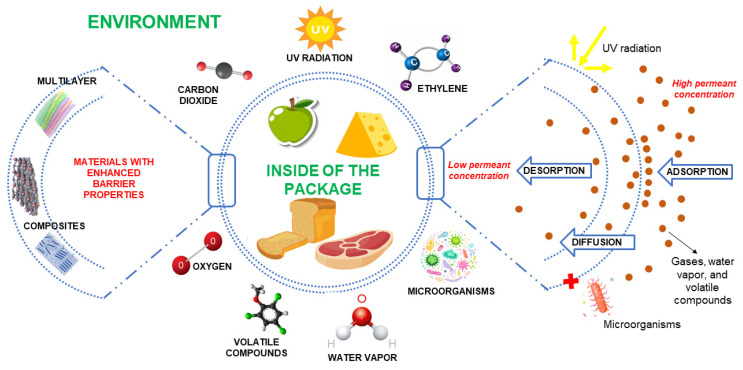
Barrier function of food packaging and hazardous agents for food safety present in the environment, along with some examples of enhanced barrier materials technologies.

**Figure 3 foods-12-01057-f003:**
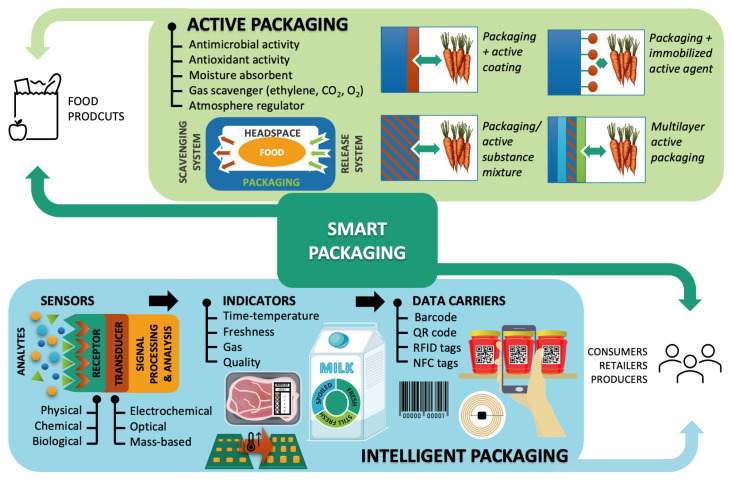
Classification of active, intelligent, and smart packaging, considering main properties, applications, interactions, and some examples.

**Figure 4 foods-12-01057-f004:**
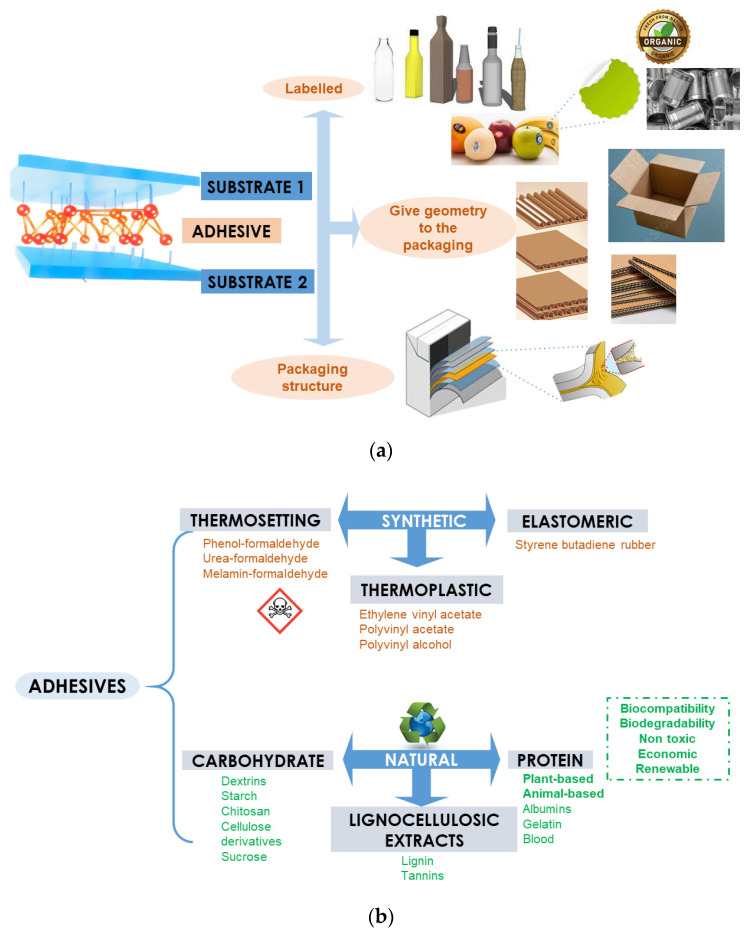
(**a**) Schematic representation of the function of adhesives in food packaging on different substrates; (**b**) Classification of adhesives according to the source of the polymers.

**Figure 5 foods-12-01057-f005:**
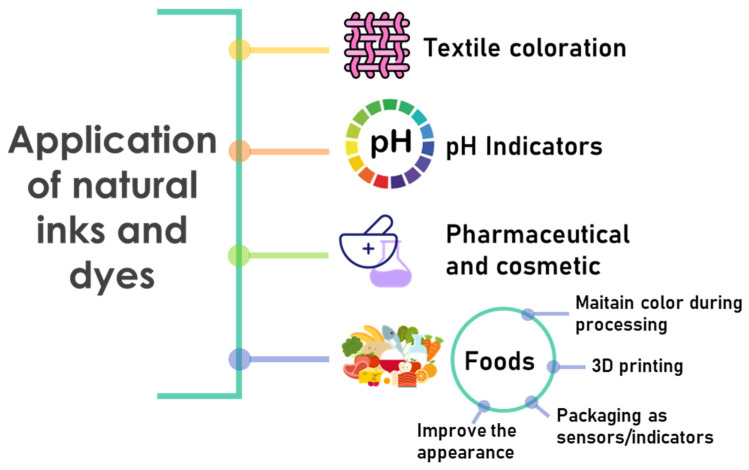
Areas of application of natural inks and dyes.

**Table 1 foods-12-01057-t001:** ASTM standard methods for the determination of gasses and water vapor permeability in polymeric food packaging.

Norm	Method	Permeant
ASTM D 1434	Manometric/volumetric	All gasses
ASTM D 7709	Gravimetric	Water vapor
ASTM E 3985	Dynamic with electrochemical sensor	O_2_
ASTM E 96	Gravimetric	Water vapor
ASTM E 2945	Static cells with analytic technique	All gasses
ASTM F 3136	Accumulation method with optical sensor	O_2_
ASTM F 1249	IR sensor	Water vapor
ASTM F 2622	Dynamic with sensor	O_2_
ASTM F 3299	Coulometric P_2_O_5_ sensor	Water vapor
ASTM F 2476	Dynamic with IR sensor	CO_2_

**Table 2 foods-12-01057-t002:** Active and intelligent biodegradable films and their food applications.

Active Biodegradable Films			
Biopolymer	Active Compound	Food Application	References
Soy protein isolate	Montmorillonite (0.5%wt) + clove essential oil (0.5%v)	Bluefin tuna filets	[253]
Chitosan and Corn starch	Lemon essential oil (1–3%wt) and grapefruit seed extract (1–3%wt)	Blueberries conservation simulating transport and commercial conditions	[182]
Corn starch	Green synthesized AgNPs in situ (143 ppm)	Cheese preservation	[173]
Tapioca starch	Grape pomace extracts (8% *v*/*v*) and cellulose nanocrystals (10% *v*/*v*)	Ready to eat chicken meats	[254]
PLA	Commercial nanoparticles: TiO_2_ (3%wt); (2%wt) nano-TiO_2_ + (1%wt) nano-Ag	Cottage cheese preservation	[255]
Curdlan + PVA	Thyme essential oil (1–2%wt)	Chilled pork meat preservation	[256]
Whey protein	Oregano and garlic essential oils (2%wt)	Kasar cheese	[257]
Chitosan-Cassava TPS bilayer films	Oregano and or cinnamon leaf essential oils (0.25%wt)	Sliced pork meat	[258]
Gelatin/Gellan gum	Red radish anthocyanins 5, 10, 15, and 20 mg/100 mL	Milk and fish quality	[259]
Zein	Laurel or rosemary leaves extracts (1–10%)	Cheese slices	[260]
Chitosan	Propionic acid	Pastry dough	[168]
**Intelligent Biodegradable Films**			
**Function**	**Intelligent System and** **Innovative Characteristic**	**Food Application**	**References**
Temperature sensor	A passive RFID tag modified with a copper-doped ionic liquid	Fresh products, for the identification of cold chain failures	[245]
Temperature abuse indicator	Au nanoparticles included in alginate hydrogel. Biobased, food safe, cost-effective, time sensible	Fresh products	[261]
Thermal insulation	Commercial pale-yellow carnauba wax. Biobased and biodegradable insulator	Beverages	[262]
pH-based freshness indicator	Biodegradable films containing anthocyanins from different sources. Real-time monitoring of food freshness.	Fresh products: cheese, yogurt, fish, pork, shrimp, and beef	[250,263,264,265,266,267]
pH sensitive	Natural compounds showing color changes with pH in biodegradable films	Fish and seafood products	[85,212,268]
CO_2_ detector	Labels containing natural (anthocyanins) or commercial (bromothymol blue and tetrabutyl-ammonium) dyes.	Fermented products such as kimchi	[269]
Oxygen indicator	UV-light activated oxygen sensitive biobased film with methyl blue indicator	Suggested for food products packed in modified atmosphere	[270]
Hydrogen sulfide indicator	Biobased films containing silver nanoparticles (detect up to 0.81 μmole H_2_S) or ferrous sulfate (detect 100 ppm H_2_S) and had a fast response (3 min).	Meat and meat products. Chicken breast and silver carp	[85,271]
Humidity indicator	Colorimetric-based sensor on photonic cellulose nanocrystals	Suggested for pharmaceutical products, cereals, and grain seeds storage	[272]

## Data Availability

Data sharing is not applicable—no new data generated.

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
