# Peer review of "Sustainable and Bio-Based Food Packaging: A Review on Past and Current Design Innovations"

_foods, 2023, doi:10.3390/foods12051057_

Round 1

Reviewer 1 Report

The title of the paper suggest focus on design innovations but there is a great deal of basic background information that distracts from the innovations.  The innovations are somewhat buried within sections that are scattered about the paper.  Also the integration of sustainable and intelligent materials has been covered before and although there are some good connections made, it is hard to see what makes this different from other papers.  I suggest completely taking out large sections that seem to inform the readers about the basics of food packaging, sustainability, active and intelligent packaging and focus strictly on unique, new innovations in the field.  For example, p.5 starting with the line 169, should be cut through p.8 line 307 (beginning with section 2.1.2.  The audience for this paper will likely know these basics and it's not needed to understand the upcoming information.  Further sections to remove would be p.9 lines 354-390.  These are just a few examples.  By removing these "background sections" then the authors can focus on the innovations and new developments.

Another point is that bio-based materials are not viewed widely as full replacements for plastic materials.   For example, compostability is not practical in many situations but have niche applications.   This is not covered in this paper.  Some places have actually put bans on compostable material because they contaminate the recycling stream.

The authors are attempting to tackle too much in this paper.  There are some new and interesting facts and information presented so that needs to be brought out but large sections of this paper are not new and distract from the purpose of the paper.

Author Response

Dear Reviewer,

We appreciate your revision of the manuscript and the suggestions given. We have conducted the revision of our manuscript following the proposed ideas. In this regard, the whole manuscript was checked and corrected for adequate English use as suggested.

In addition, based on your suggestions we have restructured the manuscript, modified Section 2, which was moved to Section 3 and extended the discussion on the sustainability of packaging materials including concerns on compostable and biodegradable materials as well as the active and intelligent packaging materials description.

We consider that the introduced modifications based on your and the other Reviewers’ comments have led to a clearer and improved paper, highlighting and focusing on the innovations and new developments in food packaging. We hope that our revised manuscript meets your expectations.

Sincerely,

The authors

Author Response

Dear Reviewer,

Thank you very much for your comments and suggestions. We are pleased to know that you found our work of interest. We have conducted the revision of our manuscript following the suggestions given and have addressed the questions raised in the revised manuscript. We list below the answer given for each comment as previously numbered indicating where they were included in the text:

Reviewer comment:

  1. Line 143, in barrier property section, the author listed the individual

importance of oxygen, carbon dioxide and moisture content. But readers

expect to read the information on that how the structure, morphology, etc.

of these gases affected the penetration rate of the packaging materials,

which is in accordance with the title.

Answer:  The following explanation was included in the manuscript: “The enhancement in the barrier properties in nanocomposite materials is attributed to the more tortuous path created by the presence of different nanofillers. This fact is explained considering that the nanofillers force the low molecular weight molecules to adopt particular tortuous pathways, producing a significant lag time. A higher aspect ratio of fillers compels the permeating gas molecule to follow a more tortuous path, leading to improved barrier properties [134]. Furthermore, gas diffusion through materials is also controlled by the crystal structure domains; it is generally assumed that ordered crystalline domains should act as an effective barrier to the diffusion of gasses and small molecules, making the amorphous phase the only pathway available for permeation. Moreover, penetrants cannot sorb in crystalline domains because their solubility coefficients are lower compared to those of their amorphous counterparts [118].” (Lines 458-468)

Reviewer comment:

  1. Line 241, in enhanced gas barrier property, explain why the particle size

or large surface area of nanoparticles like MMT affected the barrier

properties. Increased migration path? Increased crystal area? Or

something else.

Answer: The following explanation was included in the manuscript: “To achieve a better barrier performance, the assembly of nanoparticles to obtain nanocomposites constitute a strategy to improve barrier properties. Organic nanoparticles such as carbon nanotubes, nanocrystals of cellulose or starch, and inorganic nanoparticles as nanoclays or montmorillonite (MMT), can physically and chemically interact with polymeric matrices to induce stronger and reinforced structures enhancing both the barrier and mechanical properties [135]. MMT nano-clays are able to enhance the polymer barrier performance because of providing exfoliated structures that enlarge the tortuous path for small diffusing molecules due to their small particle size, high aspect ratio, and exceptionally large surface areas [134]. Thus, the nanomaterials can be applied to improve the performance of conventional materials, due to their particle size and their large surface area [136].” (Lines 469-479)

Reviewer comment:

  1. Line 306. Why Plasma affects the hydrophilicity of films? The title of this

manuscript is focused on “materials”. So in this part the author needs to

clarify how the plasma influenced the structure of the materials.

Answer: An explanation of the effects of plasma on the treated materials was included as suggested  (Lines 543-549): “The effects of plasma occur on the surface of the material without altering the properties of the bulk involving the generation of reactive species, such as ions, radicals, electrons, photons, and other excited species. Generally, the plasma induces different reactions such as surface cleansing, removal of organic contaminants, degradation (etching), cross-linking of polymer chains, and modification of the functional groups present on the surface. The different physical and chemical changes that the surface experiences depend on the gas used to generate the plasma [143].”

Reviewer comment:

  1. Line 508, typo, supply chain.

Answer: Thank you, the typo was corrected.

Reviewer comment:

  1. Line 610. Please describe the active packaging in order, either based on

function, or active agents. The current order contains repeated

information, and not acceptable.

Answer: Thanks for your suggestion. This section was extensively modified to prevent information repetition. Due to the numerous types of substances used in food packaging that can act as active agents, the classification based on the purpose of the active packaging results clearer and more useful in terms of application. However, many active compounds can have more than one function making their classification complex. Therefore a classification by group of active compounds with all the functionalities imparted to the active materials for food packaging was included. Please see (Section 4.1) on the revised version of the manuscript (Lines 655-764).

Reviewer comment:

  1. Line 735. What is the mechanism of polymer-based TTI?

Answer: Polymer-based TTIs systems are based on polymerization reactions, typically poly-diacetylene (PDA) based. PDAs present thermochromism, among other chromisms, changing from blue to red upon stimulation with various sensing applications. This information was included in the manuscript (See lines 841-844) with appropriate references.

Reviewer comment:

  1. Line 738. What is the mechanism of fading ink?

Answer: Fading inks are colored compounds that produce transparent or lightly colored products by reaction, whose kinetics are regulated by time and temperature, directly indicating changes of food shelf life. This information was included in the manuscript (See lines 850-852) with relevant references.

We think that the revised version of the manuscript is clearer and improved based on your comments. We appreciate the thorough and critical revision of our work and hope that this revised manuscript meets your expectations.

Sincerely,

The authors

Reviewer 3 Report

Good paper and dealing with a very important cocern.The different aspects of the need for a new culture of th

new culture of the food packaging are discussed but not that one of the costs during the all LC

The different new materials have different costa of production and their recycling too can be 

che key in terms of saving money.

Author Response

Dear Reviewer,

We appreciate the time given to revise our manuscript and we are pleased to know that you found our work of interest. We have revised the English of the whole manuscript for spelling and typing mistakes as suggested. Based on your comments, a further discussion on packaging costs through their life cycle was included in the Introduction and Section 2 of the manuscript.

We appreciate the thorough and critical revision of our work and hope that this revised manuscript meets your expectations.

Sincerely,

The authors

Round 2

Reviewer 1 Report

I appreciate the authors work and attempts to revise the paper.  It is better than the previous version with regard to organization but it still doesn't focus on innovations.   It is mostly a review of past work with some new/present work.  Based on that I was initially cutting large sections that were focused on basic/known information that is well covered in book chapters and other review papers in journals.  Toward the last 1/3 of the paper I realized there was so much being cut that would leave very little to publish.  Therefore, I recommend you change the title of the paper to  a Review of past and present innovations in sustainable bio-based food packaging.  I would still leave out the active and intelligent packaging section because the areas of bio-based sustainable packaging and active and intelligent are too different to be integrated into one paper.

I knew there was a tight deadline for getting this back so I didn't finish revisions because it will be based on whether the authors are interested in changing the title or not.  I encourage the authors to consider this change as I believe their hard work is worthy of publishing, but just not with the title of Innovations because the focus so strongly focuses on well known technologies.

Author Response

Dear Reviewer,

Thank you very much for this second revision, we appreciate that the changes introduced based on the previous revision render a more organized and clear manuscript. Based on your suggestion we have modified the title of the review so that it reflects better on the work content: Sustainable and bio-based food packaging: a review on past and current design innovations”. Besides, we have introduced the corrections and updated some of the information as requested by the reviewer throughout the text. Moreover, we agree that the review involved a wide range of topics on food packaging that is difficult to encompass on a single work. However, the idea of including active and intelligent in a separate section was to contextualize their role of on sustainable packaging and highlight some current work on th field. In this regard, a careful revision on the published works on biobased packaging materials was conducted including relevant and current references. Lastly, there is some information in these sections that is included as introductory to the topic even though the authors are aware that it has been reported elsewhere.

We appreciate the thorough and critical revision of our work and hope that this revised manuscript meets your expectations.

Sincerely,

The authors

Reviewer 2 Report

none

Author Response

Dear Reviewer,

We are glad the revised manuscript met your expectations, and we are thankful for your revision work.

Sincerely,

The authors